# Adaptive Antenna for Maritime LoRaWAN: A Systematic Review on Performance, Energy Efficiency, and Environmental Resilience

**DOI:** 10.3390/s25196110

**Published:** 2025-10-03

**Authors:** Martine Lyimo, Bonny Mgawe, Judith Leo, Mussa Dida, Kisangiri Michael

**Affiliations:** School of Computational and Communication Science and Engineering (CoCSE), The Nelson Mandela African Institution of Science and Technology (NM-AIST), Arusha 2331, Tanzania; bonny.mgawe@nm-aist.ac.tz (B.M.); judith.leo@nm-aist.ac.tz (J.L.); mussa.ally@nm-aist.ac.tz (M.D.); kisangiri.michael@nm-aist.ac.tz (K.M.)

**Keywords:** adaptive antennas, beamforming, energy efficiency, IoT, LoRaWAN, SNR, maritime environment, systematic review

## Abstract

Long Range Wide Area Network (LoRaWAN) has become an attractive option for maritime communication because it is low-cost, long-range, and energy-efficient. Yet its performance at sea is often limited by fading, interference, and the strict energy budgets of maritime Internet of Things (IoT) devices. This review, prepared in line with the Preferred Reporting Items for Systematic Reviews and Meta-Analyses (PRISMA) 2020 guidelines, examines 23 peer-reviewed studies published between 2019 and 2025 that explore adaptive antenna solutions for LoRaWAN in marine environments. The work covered four main categories: switched-beam, phased array, reconfigurable, and Artificial Intelligence or Machine Learning (AI/ML)-enabled antennas. Results across studies show that adaptive approaches improve gain, beam agility, and signal reliability even under unstable conditions. Switched-beam antennas dominate the literature (45%), followed by phased arrays (30%), reconfigurable designs (20%), and AI/ML-enabled systems (5%). Unlike previous reviews, this study emphasizes maritime propagation, environmental resilience, and energy use. Despite encouraging results in signal-to-noise ratio (SNR), packet delivery, and coverage range, clear gaps remain in protocol-level integration, lightweight AI for constrained nodes, and large-scale trials at sea. Research on reconfigurable intelligent surfaces (RIS) in maritime environments remains limited. However, these technologies could play an important role in enhancing spectral efficiency, coverage, and the scalability of maritime IoT networks.

## 1. Introduction

Communication in marine environments is possible through various technologies, including Satellite, Cellular, Acoustic, LoRaWAN, Underwater or Sonar communication, etc. Among these technologies, LoRaWAN technology is known for its low-cost and long-range communication [1]. Wireless communication in marine environments faces many challenges that cause the coverage distance to be short [2]. These obstructions arise from several factors, including multipath propagation caused by reflections from water surfaces and maritime structures [3], leading to intersymbol interference (ISI), where overlapping signal echoes from multiple paths distort the received symbols and reduce communication performance [4]. ISI reduces SNR, raises the bit error rate (BER), and weakens the signal. Adaptive antennas have played a key role in advancing wireless communication, improving signal quality and extending coverage [5]. These antennas adjust their transmit and receive patterns in real time, directing beams toward users and reducing interference to improve performance [6]. This performance is achieved through identifying the direction of incoming signals, enabling more effective communication in tough environments [7].

LoRaWAN builds on the LoRa physical layer, which uses Chirp Spread Spectrum (CSS) modulation to achieve the long-range, low-power characteristics needed in Low Power Wide Area Network (LPWAN) systems [8]. Because it can sustain energy-efficient links across large distances, LoRaWAN has become a strong candidate for maritime applications where coverage and power constraints are critical [9]. One of the key limitations lies in the antenna side; most LoRaWAN antennas are designed with terrestrial deployments in mind. They are not optimized for the rapidly changing conditions at sea. This means that antenna design choices often shape the actual coverage range, while interference from other devices can further degrade performance in maritime settings [10].

Traditional fixed-beam antennas deployed in marine environments regularly find it difficult to adapt to the varying conditions, thus making it hard to control interference and maintain good coverage [11,12,13]. However, several studies have been carried out on adaptive antenna technologies for earth-based applications; little work has been carried out to adapt and optimize them for maritime settings, especially in LoRaWAN systems [14,15,16]. LoRaWAN, through LoRa modulation, allows for wide coverage and robustness against interference, with a span of 5 km in urban areas and up to 15 km in rural areas [17]. Its low operating speeds (0.3 kbps to 50 kbps) make it well-suited for battery-powered sensors and trackers [18,19]. Adaptive antennas can ease these challenges by directing signals and reducing interference rather than relying on fixed patterns. In maritime LoRaWAN, their use is limited and scattered compared to terrestrial systems.

The requirements of maritime IoT are quite different from those of land-based networks. LoRaWAN applications in maritime settings, such as vessel tracking, environmental monitoring, and emergency alerts, must balance continuous and event-based transmissions while accounting for acknowledgements, latency, and payload limits. For example, a vessel tracking system must send frequent position updates with minimal delay. By contrast, environmental monitoring can be scheduled or event-driven to save energy. Among all traffic types, the most demanding are alerts, as they rely on highly reliable delivery supported by acknowledgments to ensure the message is received. These needs determine how frequently devices transmit, how much delay is acceptable, and the payload size, which in LoRaWAN ranges from 51 to 243 bytes depending on the spreading factor (SF) [20,21,22,23].

This review looks at how adaptive antenna designs could be uniquely applied to maritime LoRaWAN. We grouped existing antennas into switched-beam, phased-array, reconfigurable, and AI/ML-enabled approaches, then examined their reported benefits for coverage, reliability, and energy efficiency in open-water conditions. The review also has the following contributions. (i) It examines propagation challenges and adaptive antenna solutions in maritime LoRaWAN environments; (ii) it classifies antenna types and evaluates their strengths and limitations; (iii) it synthesizes maritime application requirements; and (iv) it highlights open challenges and future research directions.

The paper is structured in seven sections. Section 2 introduces LoRaWAN and explains why its features matter in maritime applications. Section 3 describes the review method. Section 4 looks at the main propagation challenges together with antenna fundamentals, while Section 5 discusses design options and their integration into LoRaWAN systems. Section 6 brings the results together in relation to the research questions, and Section 7 closes with open problems and directions for future work.

## 2. Background on Maritime LoRaWAN

### 2.1. LoRaWAN Architecture Overview

LoRaWAN is a low-power networking protocol that runs in unlicensed sub-GHz bands and relies on spread spectrum modulation [24]. It supports six spreading factors, which allow a balance between range and data rate from a few hundred bits per second up to around 50 kbps, while each message can carry as many as 243 bytes. A practical advantage of LoRaWAN is that transmissions from one device can be heard by several gateways at the same time, adding redundancy and improving reliability in demanding environments [25]. The protocol defines three device classes to meet different application needs. Class A is the most energy-conscious; devices only open a receive window after sending, which works well for event-driven uses such as water-quality buoys or emergency alerts. Class B introduces scheduled listening periods, reducing latency and making it more useful for coordination tasks like fleet management. Class C keeps the receiver permanently open, consuming more power but enabling real-time services such as ship tracking, collision avoidance, or remote control [25,26].

LoRaWAN networks follow a star-of-stars architecture. End devices placed on ships, buoys, or coastal stations collect data such as tide levels, vessel positions, or fishing activity and send it with very low energy use [27]. Gateways capture these signals and pass them to a central Network Server, which handles duplication checks, rate adaptation, encryption, and other network tasks [28]. An application server then turns the data into usable information for maritime monitoring and control [29]. With this layered setup, LoRaWAN can deliver scalable, power-efficient connectivity suitable for long-term maritime IoT deployments. Figure 1 shows a maritime LoRaWAN network, where end devices—including container trackers, weather and water quality sensors, aquaculture monitors, and vessel trackers—communicate through shore-based, ship, port authority, and offshore buoy gateways. These gateways forward the data to a network server, which then routes it via the internet to the application server for processing and analysis.

### 2.2. Related Studies

Research on LoRaWAN has grown quickly, yet most surveys still treat it from a general IoT angle rather than from the sea. Banti et al. [30] provided an overview of LoRaWAN protocols with a particular emphasis on energy efficiency. In contrast, Alkhayyal and Mostafa [23] examined the use of AI and ML to optimize network performance. These are useful contributions, but they remain grounded in terrestrial use cases and pay little attention to the physical realities of the ocean.

Some application-oriented studies have taken LoRaWAN offshore. Parri et al. [31] tested it in the Sea Factory project for aquaculture and confirmed long-range links, though antenna adaptability was not part of the analysis. Rohde et al. [32] pushed the technology to Antarctica for sea-ice monitoring, where range limits became obvious, but again, antenna strategies were left aside. Elgharbi et al. [33] tackled reliability through a mesh-based design, improving resilience but still leaving propagation issues such as multipath fading or Doppler shifts unsolved.

What ties these earlier efforts together is the lack of a systematic look at adaptive antennas in maritime LoRaWAN. Unlike protocol-driven or routing-focused work, this review brings together results on switched-beam, phased array, reconfigurable, and AI-assisted antennas. By setting these approaches against specific maritime challenges, mobility, reflections from the sea surface, and persistent interference, it opens a clearer view of how adaptive antennas could help LoRaWAN scale and operate more robustly in open-water conditions.

### 2.3. Multi-Gateway Reception in LoRaWAN

A distinctive strength of LoRaWAN is its ability to have uplink messages received by several gateways at once. In practice, this means that if one gateway fails to catch a packet because of interference or fading, another can often pick it up, giving the system a built-in safety net [34]. This form of diversity is especially useful at sea, where reflections from waves and fast-changing channel conditions frequently undermine single links.

Field evidence points to the same conclusion. Pensieri et al. [35] showed that long-distance connections, sometimes extending hundreds of kilometers, were possible when several gateways were deployed along the coast. In another study, Sagala et al. [36] used three gateways for real-time vessel tracking and still kept delivery ratios above 97%. Lahoud and Khawam [37] later introduced a macro-diversity method, where packet fragments collected by different gateways are combined to improve reliability and capacity. These examples show that in maritime LoRaWAN systems, having multiple gateways receive the same transmission is not simply an advantage but a necessity for ensuring stable and reliable connectivity.

### 2.4. Maritime-Specific Requirements

Maritime LoRaWAN networks face challenges that rarely appear on land. Moving vessels and drifting buoys constantly reshape wireless links and cause Doppler shifts, so stable connections often depend on antennas that can adjust their beams in real time [35,38]. Power is another constraint where most devices rely on batteries or renewables, so hardware and protocols must save energy without sacrificing reliability [39]. Coverage is also limited; in many cases, only one offshore or coastal gateway is available, which makes high-gain antennas and careful link budgeting essential for long-range operation [40]. At the same time, salt spray, humidity, and severe weather demand marine-grade enclosures to protect equipment [41,42]. Spectrum rules further restrict duty cycles and available bands, which forces efficient spectrum use and adaptive data rates [43]. On top of this, reflections from the sea surface cause multipath fading and channel fluctuations. Adaptive antennas help counter this by suppressing interference and adding spatial diversity [44,45].

A further complication is that some applications must link devices across different media. LoRaWAN works well in the air and over water, but signals fade quickly below the surface, which prevents direct underwater communication [46,47]. Hybrid solutions through mixing RF with acoustic or optical links, or even using Unmanned Aerial Vehicles (UAVs) as relays, are now being tested to bridge the gap between surface nodes and underwater assets [13]. Studies agree that handling these cross-media transitions will be key to building robust and scalable maritime IoT systems [48].

### 2.5. LoRaWAN Network Configuration

This study explicitly considers a LoRaWAN deployment rather than isolated point-to-point LoRa links. The network is assumed to operate primarily in the European Union (EU) 868 MHz band, while also discussing the 433 MHz band as an alternative for extended maritime coverage in line with the LoRaWAN specification. End devices are modeled mainly as Class A nodes to reflect energy-constrained maritime sensors, with references to Class B and Class C operation in cases where lower latency or continuous reception is required. To capture the inherent multi-gateway reception feature of LoRaWAN, at least two gateways are considered, representing a coastal station and an offshore platform.

In the EU868 MHz band, duty-cycle limits range between 0.1% and 10% depending on the sub-band [22,43], and must be taken into account when designing maritime LoRaWAN deployments to balance compliance and throughput. As a result, the network traffic is mostly uplink-oriented; ships, buoys, or offshore platforms send telemetry and sensor data, while downlinks are kept minimal, often just acknowledgments or urgent control messages.

Deployment scale is another factor that shapes antenna design. For a single buoy or vessel operating far offshore, the priority is to reach the gateway, so high-gain directional antennas are often the best fit. But in crowded zones such as fishing areas, ports, or along coastal infrastructure, the problem shifts. Here, interference between many nodes can reduce reliability, and adaptive or beam-steering antennas become more effective in keeping the network stable. Past studies on LoRaWAN scalability show that when the number of devices rises, packet delivery ratios drop unless new gateways are carefully placed to share the load [18,49]. This makes it clear that antenna strategies must account not only for range but also for user density.

A comparative analysis of various technologies related to LoRaWAN, including their key characteristics, strengths, and limitations within maritime application contexts, is presented in Table 1.

### 2.6. Maritime Propagation Characteristics and Models

Maritime propagation channels behave very differently from those on land because signals are shaped mainly by line-of-sight (LoS) paths, sea-surface reflections, and changing atmospheric conditions. One of the most significant effects comes from the ocean surface itself. Reflections over the sea form a two-ray channel, where the direct and surface-reflected signals may combine constructively or destructively depending on antenna height, phase, and distance. The outcome is often severe fading, making the connection highly unstable [61,62,63,64]. Although open water typically shows lower Free-Space Path Loss (FSPL) than cluttered land environments, the channel conditions remain highly variable and rarely stable. Tides, fog, humidity, and rainfall all alter propagation in ways that require adaptive strategies to maintain performance [65,66,67]. Because LoS dominates maritime links, even short interruptions from large vessels or onboard structures can degrade signal quality, which is particularly critical for narrowband technologies like LoRa [16,35]. At the same time, atmospheric refraction and tropospheric ducting—driven by temperature and humidity gradients—can on occasion extend LoRaWAN coverage to hundreds of kilometers. Still, these effects also introduce sudden and unpredictable fluctuations in link reliability [68,69].

Several models are employed to represent these maritime propagation effects. The FSPL model provides a baseline estimate but neglects multipath and atmospheric factors [70]. The FSPL equation is given in (1).(1)$$FSPL\left(dB\right)=20{log}_{10}\left(d\right)+20{log}_{10}\left(f\right)+32.44$$
where d is distance in kilometers (Km), and f is frequency (MHz).

The FSPL model at 868 MHz is illustrated in Figure 2, which shows that the signal falls off logarithmically with distance rather than quadratic variation, as predicted by the inverse square law.

The Two-Ray model explicitly captures sea-surface reflections but is less accurate under rough-sea conditions [62,71]. It is based on a far-field assumption and destructive interference between the direct and reflected rays, from which a greater loss for free space is demonstrated in (2), where h_t_ and h_r_ are the heights of transmitter and receiver antennas.(2)$$PL\left(d\right)≅40{log}_{10}\left(d\right)-20{log}_{10}\left(h_{t}h_{r}\right)$$

Figure 3 shows the maritime two-ray propagation model, which takes into account both the direct path and reflected path over the sea, demonstrating the effect of the difference in path length and the interference with the reflection on the water surface.

In maritime communication, several propagation models are commonly applied, each suited to different conditions. The ITU-R P.1546 model is the standard choice for long-range sea coverage, though it often needs local calibration to capture site-specific variations [72]. In harbors or areas with obstructions like cliffs and vessel superstructures, knife-edge diffraction models better represent shadowing effects [72,73]. Weather-driven anomalies such as tropospheric ducting are instead captured by refractive models, which predict coverage extensions caused by temperature and humidity gradients [74] or more localized environments, including fjords, archipelagos, or offshore platforms. Empirical and measurement-based models provide accurate predictions. However, their use is limited outside the measurement region [75]. Table 2 provides an overview of these models, their typical applications, and their constraints.

### 2.7. Adaptive Antennas

Adaptive antennas improve performance by dynamically steering their radiation patterns in real time, allowing them to optimize gain, control beamwidth, and suppress interference [77,78,79]. This adaptability is especially important for LoRaWAN in maritime environments, where rapidly changing propagation conditions require reliable and energy-efficient links [80]. Four categories of adaptive antennas studied in recent literature regarding maritime LoRaWAN applications are discussed in this section.

#### 2.7.1. Switched-Beam Antenna

Switched-beam antennas use a fixed number of distinct patterns, or beams, to achieve directionality [81], making them a feasible solution for maritime LoRaWAN gateways built into buoys or on ships, where energy consumption limits the use of more complex systems. These antennas provide intermediate beam steering with less complexity as compared to the continuous adaptation and potentially better coverage on open-sea point-to-point communication links [82]. Switched-beam antennas can be a fast steering and high-gain solution with small size, weight, and power for some LoS hotspots and to avoid complex mechanically pointed and expensive phased array antennas [83]. Switched-beam antennas deliver moderate directional improvements and remain highly energy-efficient because of their low-complexity switching logic. When integrated with sealed enclosures, they offer good resilience against maritime environmental conditions [84].

#### 2.7.2. Phased Array Antenna

Phased arrays use electronically controlled phase shifts to steer beams dynamically, avoiding mechanical movement [85]. This feature is especially relevant for coastal LoRaWAN gateways and marine vessels that need to track moving end-nodes in real time, including autonomous surface sensors and fishing boats. Since phased array antennas are highly directive and reconfigurable, they can strengthen signals and reduce packet loss, even when communicating across long stretches of ocean. For example, phased arrays have been shown to improve both the Received Signal Strength Indicator (RSSI) and the reliability of LoRa links at ranges exceeding 10 km for marine environments [35]. Phased arrays provide precise beam steering, deliver strong SNR performance, and can dynamically track moving nodes in maritime environments. However, their high energy consumption and sensitivity to mechanical stress make them better suited for stable, high-power platforms like ships or coastal gateways [86].

#### 2.7.3. Reconfigurable Antennas

Reconfigurable antennas can change their electrical or physical features such as frequency, radiation pattern or polarization, using elements such as varactors, Micro-Electro-Mechanical Systems (MEMS), or P-type Intrinsic N-type Diode (PIN diode), which provide a trade-off between flexibility and power consumption [87,88,89], making them especially well-suited for smart buoys and marine IoT systems that need to deal with interference-prone or bandwidth-constrained environments. Studies have also proved bands of 868 MHz and 915 MHz LoRaWAN frequency-reconfigurable antennas, which can deliver flexibility in adaptation to regional band standards [90]. Marine sensor nodes with pattern reconfigurability enhance link robustness, while reconfigurable antennas adapt frequency or pattern to changing conditions [91].

#### 2.7.4. Smart Antennas with AI/ML Control

AI/ML-assisted smart antennas can further offer the possibility of dynamic optimization of beam patterns, orientation, and other characteristics in real time, given environmental feedback [92,93]. These systems are emerging in maritime LoRaWAN installations where sea conditions, weather, and mobile nodes create much more variation. For instance, in [94], reinforcement learning is employed for adaptive parameter modulation of antennas, which achieves higher throughput and lower energy consumption in marine environments with varying wave conditions. Moreover, AI-based reconfigurable antennas have been surveyed, and it is mentioned that machine learning provides a high level of autonomous adaptation in the time-varying and shadowing maritime scene, particularly in the context of uncrewed marine systems and intelligent navigation systems [23]. Although they are promising in terms of performance, they are still energy-intensive and are currently viable in advanced or experimental maritime systems [92].

#### 2.7.5. Adaptive Antenna Integration in LoRaWAN

Adaptive antennas can be incorporated into LoRaWAN networks in several forms, depending on their complexity and control requirements. A switched-beam module may be attached directly to an end device, where the microcontroller unit (MCU) selects the active beam based on link-quality indicators such as RSSI or SNR reported by the transceiver [84]. This lightweight approach is attractive for maritime nodes like buoys, where energy supply is limited. In contrast, phased-array and AI/ML-based antennas demand higher processing power and are usually managed at the gateway [6,86]. A dedicated controller executes beam steering and pattern reconfiguration, while the network server can provide coordination or optimization policies [14].

The outcome is a hierarchical control structure: basic beam switching handled at the device, and advanced adaptation executed at the gateway with possible input from the server. This arrangement reduces the burden on energy-constrained nodes yet still enables the network to benefit from improved coverage, interference management, and reliability in dynamic maritime conditions [31,49]. Figure 4 illustrates this architecture; simple switched-beam antennas are controlled by the end device through its MCU, whereas more advanced phased-array and AI/ML-based designs are handled at the gateway and can be coordinated with the network server. Data paths are shown with solid arrows, while dashed arrows indicate control and feedback signals.

### 2.8. Reconfigurable Intelligent Surfaces (RISs) for Maritime LoRaWAN

RIS is gaining attention as a lightweight way to improve wireless links by controlling how signals reflect and propagate. In maritime LoRaWAN systems, RIS can play an important role by softening the impact of multipath fading, reducing reflections from the sea surface, and easing shadowing caused by ships or coastal structures, all of which often weaken links at sea. Recent studies have even suggested dual-RIS setups to connect different transmission domains, including air, surface, and underwater, opening new opportunities for integrated maritime networks [95]. Other work shows that RIS can lower bit error rates without needing full channel state information, which is valuable in fast-changing sea conditions where Channel State Information (CSI) is difficult to track [96]. Recent work has also focused on energy-harvesting RIS panels for offshore platforms, enabling them to operate autonomously while supporting low-power IoT devices [97]. Rather than replacing adaptive antennas, RIS should be seen as a complementary technology that strengthens their effectiveness and helps build maritime LoRaWAN networks that are more robust and scalable.

### 2.9. Design Trade-Offs of Adaptive Antenna Techniques in Maritime LoRaWAN Networks

Signal strength improvement, coverage, and interference management can be achieved through adaptive antennas, but they also add complexity to the system [98]. Approaches such as beamforming and reconfigurability often require extra hardware and power, which is a drawback for maritime IoT devices with limited resources [93,99]. As shown in Table 3, comparing adaptive designs with simpler options like omnidirectional and Multiple-Input Multiple-Output (MIMO) antennas helps to illustrate the performance–complexity trade-off in LoRaWAN applications [100].

## 3. Methodology

This systematic literature review has followed the PRISMA 2020 [102] guidelines to investigate the recent development in adaptive antennas and LoRaWAN technology broadly. A total of 1605 records were identified after an extensive search in IEEE Xplore (n = 550), Scopus (n = 209) and Google Scholar (n = 846). After removing duplicates and excluding irrelevant articles (n = 1293), 312 records remained for screening. Of these, 248 studies were excluded through title and abstract screening. The full-text analysis was performed for 64 papers, of which 41 did not meet the inclusion criteria for adaptive antenna systems and maritime LoRaWAN. In the end, 23 studies were included in the systematic review.

### 3.1. Motivation and Objectives of Systematic Review

This review looks at the latest developments in adaptive antennas for maritime LoRaWAN, bringing together what is known about their designs, performance, deployment challenges, and the new technologies helping to make long-range wireless links more reliable at sea.

### 3.2. Scope of Study and Research Questions

This review focuses on performance, energy, environmental resilience-based antenna designs, operational features, the challenges for deployment, and recent highlights for long-distance maritime communication, where continuous and reliable service under harsh conditions is crucial. To facilitate such an analysis, several research questions (RQs) were developed:RQ1: What are the adaptive antenna techniques that contribute most to the energy efficiency and performance in maritime LoRaWAN?RQ2: What are the effects of marine environmental conditions on adaptive antenna response?RQ3: What are the key antenna parameters that affect reliable maritime LoRaWAN communication?RQ4: Which antenna designs are the best in terms of performance, energy efficiency, and complexity?RQ5: What research gaps exist, and in which directions should future studies be directed?

### 3.3. Eligibility Criteria

Inclusion and exclusion rules were set in advance to guide the selection process.

#### 3.3.1. Inclusion Criteria

Inclusion criteria were defined to make sure the systematic review covered only relevant studies. Articles selected for research had to focus on the technology LoRaWAN in combination with adaptive antenna systems. The selection has included studies showing the interfacing to the antenna system of LoRaWAN or any other LPWAN technology. To maintain the quality and relevance of this systematic review, the following inclusion criteria were employed:The research has to include LoRaWAN with adaptive, smart, beamforming, or reconfigurable antenna systems.Investigations should address the development, deployment, and testing of these antenna technologies in marine-relevant environments and scenarios such as offshore, coastal, shipboard, and open-water scenarios where over-water propagation properties are significant.Papers related to LPWAN and with a focus on antenna adaptation were additionally taken into account if the connection was evident in a maritime scenario.Studies that investigated general LoRaWAN and adaptive antenna were included if they provided transferable insights into maritime deployments.Publications should have been published between 2019 and 2025 in peer-reviewed journals or well-known conference proceedings.

#### 3.3.2. Exclusion Criteria

To be certain that only studies that were directly related to the scope of the research were included, the exclusion criteria were formulated as follows:Works that only dealt with LoRaWAN or LPWAN technologies without mentioning or integrating adaptive, reconfigurable or smart antenna systems.Papers with abstracts that are irrelevant or nontechnical and that do not contain enough information about antenna design, performance, or implementation in the maritime field.Chapters, books, editorial notes, white papers, and patents.Research papers that focus solely on terrestrial applications and have no indication of any relevance/transferability to design considerations for maritime, offshore, or aquatic deployment.Papers elsewhere in the range of 2019 to May 2025.

These exclusion criteria were implemented consistently at the level of abstract screening and full-text review phase to ensure the final included papers were clear, relevant and strongly in line with the scope of the systematic review.

Several studies met the initial inclusion criteria, especially those concerning basic LPWAN performance or terrestrial LoRaWAN antenna design. Still, they were excluded from the full-text screening as they did not discuss the adaptive antennas in maritime.

### 3.4. Information Sources

A literature search was performed across multiple electronic databases. The major sources were IEEE Xplore, Scopus, and Google Scholar. The search has been conducted from 15 April to 31 May 2025, for publications between January 2019 and 31 May 2025.

### 3.5. Research Strategy

A structured search was used to find literature relevant to this review. IEEE Xplore, Google Scholar, and Scopus were selected for their broad coverage of peer-reviewed journals, conference papers, and technical studies on wireless communication and antennas.

The following Boolean search query was performed: (“LoRaWAN” OR “LoRa”) AND (“adaptive antenna” OR “smart antenna” OR “beamforming” OR “switched beam” OR “reconfigurable antenna”) AND (“marine” OR “maritime” OR “sea” OR “offshore” OR “ocean” OR “vessel” OR” ship” OR “coastal”). For filtering purposes, publications were restricted to those from 2019 to May 2025 and written in English, and to peer-reviewed articles and conference papers. The final step was to screen and review the full-text articles of the included studies based on the established inclusion and exclusion criteria.

### 3.6. Selection Process

Records were exported to Mendeley Reference Manager to remove duplicates during the selection process. Two reviewers independently performed the screening at three phases. Titles were first scanned to exclude any papers not relevant to LoRaWAN or adaptive antennas, or the maritime/marine environment. Second was the screening of abstracts for relevance to the research questions, and exclusions based on publication year before 2019 or after 2025 and by language (non-English studies). Lastly, full texts of the remaining studies were reviewed to check if they meet the inclusion criteria, with a focus on those that describe adaptive antenna methods in LoRaWAN maritime scenarios. The discrepancies between the two reviewers were resolved through discussion and the achievement of a consensus. The PRISMA flow diagram in Figure 5 shows the number of screened, excluded, and included paper at each stage.

### 3.7. Data Collection Process

Two independent reviewers manually extracted data from each eligible study. Extraction was guided by a predesigned template in Microsoft Excel that allowed us to archive detailed study year, antenna type deployed, deployment location and marine context, as well as any performance metrics reported, such as gain, SNR, and power consumption. When the two reviewers reached different conclusions, these were discussed until agreement was achieved. Data extraction was carried out manually, with no automated tools used and no additional information requested from the study authors.

### 3.8. Study Risk of Bias Assessment

The review did not use a formal risk-of-bias tool. However, potential biases such as selective reporting, narrow testing conditions, and lack of result replication were identified qualitatively on data extraction and taken into account in the interpretation of findings.

### 3.9. Study Characteristics

Table 4 presents a summary of studies included in the systematic review based on the year of publication, research focus (performance, energy efficiency, environmental resilience) and/or their combinations

### 3.10. Data Items

Predefined data items from each eligible study that included publication year, antenna type, deployment type, marine setting and reported performance metrics like gain, SNR, and power consumption. Two authors independently performed data extraction, and disagreements were resolved through discussion.

### 3.11. Risk of Bias in Studies

No formal risk-of-bias tool was applied, as included studies were mainly simulation- or design-focused; potential biases were qualitatively noted, including limited validation and small sample sizes.

### 3.12. Effect Measures

This review did not conduct a meta-analysis; hence, there are no pooled statistical effect measures available to report. Instead, performance metrics such as antenna gain, SNR enhancement and power consumption were pulled from each study, and all results are stated in their units of measure for qualitative comparison.

### 3.13. Synthesis Methods

Suitable studies were classified according to adaptive antenna type, LoRaWAN application and maritime scenario to verify their eligibility for synthesis. Summary tables were created to tabulate the data, and themes with supporting figures were used for visualization. If needed, performance metrics were not provided, and they were labeled as not reported. There was no attempt to convert statistics. Since study designs, metrics and reporting formats were heterogeneous between studies, a narrative synthesis was presented rather than conducting a quantitative meta-analysis. Narrative heterogeneity was addressed by grouping findings according to the theme they represented and conducting no sensitivity analyses due to a lack of quantitative data suitable for comparison.

### 3.14. Reporting Bias Assessment

Reporting bias was not assessed formally, as this is a qualitative synthesis and the studies were fairly heterogeneous. The risk of bias was minimized through a systematic search throughout relevant databases, application of predefined inclusion or exclusion criteria, and independent screening by two reviewers.

### 3.15. Certainty Assessment

The certainty of evidence was not graded using formal frameworks due to the methodological diversity and qualitative synthesis approach implemented in this review. Instead, we used a transparent description of the study characteristics, consistently applied eligibility criteria where appropriate, and critically appraised methodological strengths and limitations across all the studies to have confidence in our findings.

### 3.16. Results

A total of 1605 records were identified from searches on IEEE Xplore, Scopus and Google Scholar. We screened 64 full-text articles for eligibility after duplicate removal, and 23 studies met the inclusion criteria. Figure 5 shows the PRISMA flow diagram of the study selection process, including the number of records identified, screened, eligible, and included. It illustrates the search process workflow, along with systematized database search, duplication deletion, and eligibility examination. The effect of the chosen studies is discussed in depth in a later stage.

### 3.17. Systematic Review Contributions

Key contributions include:Classification of adaptive antenna systems according to their use at the gateway and end-node layers, and discussion of their impact on maritime services.An investigation into how marine factors like multipath propagation, ship motion, and atmospheric loss affect antenna performance and signal reliability.The interrelation of adaptive antenna systems with LoRaWAN protocol layers is studied, specifically focusing on link quality, scalability, and system performance.A comparison is performed between gateway and end-node realizations concerning power consumption, design complexity, and computational demands.Based on the review, the primary challenges and future work, such as intelligent control, miniaturization, and environment adaptation on the next-generation maritime IoT networks, are highlighted.

Figure 6 represents the number of included studies on adaptive antennas and LoRaWAN (with relevance to maritime applications) from 2019 to May 2025. The data indicate a rising research interest that reached its peak in 2022, followed by consistent contributions in subsequent years.

## 4. Properties and Characteristics of Adaptive Antennas in Maritime LoRaWAN

This section covers the features that help adaptive antennas improve signal reliability, connectivity, and energy efficiency at sea.

### 4.1. Principles of Maritime Adaptive Antennas

Maritime adaptive antennas rely on real-time beam reconfiguration to maintain stable communication despite vessel motion, sea-surface reflections, and varying link geometries [113,114]. Using sensor fusion and adaptive control, they can dynamically alternate between narrow high-gain beams for long-range connectivity and wider beams for shorter links, while also suppressing interference through spatial filtering and null steering [115,116,117].

At the same time, reliable operation in harsh marine environments requires robust construction. Antennas are typically built with corrosion-resistant materials, protective coatings, and sealed enclosures to withstand saltwater exposure, waves, and severe weather [118,119,120]. Shock-absorbing frames and damping mechanisms are incorporated into the mechanical structure to control resonance and avoid damage during heavy sea states [121].

### 4.2. Beamforming Techniques for Maritime Applications

Beamforming at sea is shaped by two challenges that rarely occur together on land: how signals behave over open water and the constant movement of vessels. One response has been to direct the beam tightly along the horizon, which extends coverage over long distances [122]. These work to a point, but they do little against reflections from the sea surface. To address that, multipath-adaptive beamforming is often applied, reinforcing the line-of-sight path while reducing interference from echoes [117,123]. Maintaining precise beam alignment is difficult because vessel motion constantly alters the link geometry. Elevation-compensating beamforming attempts to overcome this by adjusting the beam’s vertical angle through sensors or gyroscopes. Without this correction, rough seas can quickly lead to frequent dropouts and unstable links [124].

### 4.3. Implementation Architectures

There are four fundamental implementation architectures for marine adaptive antenna systems, each capable of supporting different levels of beam control and developed for specific applications and vessel configurations.

#### 4.3.1. Phased Array Systems

The optimal design and control of phased array systems (PAS) rely on the concept of electronic beam steering by phase shift of signals among the antenna components for enhanced directivity and SNR of the signal with no mechanical movement. This characteristic enables rapid beam scanning and effective interference mitigation, both of which are essential for reliable wireless communications [125].

Figure 7 shows an example of a phased array antenna system, where several antenna elements receive or transmit signals with variable phase shift and attenuation of signals to steer signals by detecting constructive or destructive interference of the signal to a direction of interest. The radiation characteristic of an N-element antenna array is spatially scanned, and the directivity is increased by controlling the amplitude and phase of each antenna unit. For N identical antenna elements, the array model falls into two main constituents: (1) the element pattern, which describes the radiation pattern of each antenna; (2) the array factor (AF), which represents the radiation pattern of the N-element array with the antennas replaced by isotropic elements. Hence, the composite pattern is obtained by multiplying the element pattern by the AF [126]. The normalized array factor of the antenna array is expressed as(3)f=1NsinN2−2πdλsinθ+φPSsin12−2πdλsinθ+φPS
where *N* is the total number of the array elements, *d* is the separation distance between the adjacent array antennas, *θ* is the angle of the incident or departure waves to the normal plane of the antenna surface, and *φ*PS is the phase shift between two adjacent antennas given by the phase shifter. Equation (3) indicates that the AF reaches its maximum when(4)$$\varphi_{PS}=\frac{2\pi d}{\lambda }\sin\theta$$

Therefore, the signals are constructively added at angle θ for the receive or transmit mode, but are suppressed along other directions.

#### 4.3.2. Parasitic Array Antennas

Parasitic array antennas are compact, broadband designs that use a simple parallel-strip line feed, eliminating the need for baluns while maintaining impedance matching and suppressing unwanted currents [127]. By using PIN diode-controlled parasitic elements, these antennas can rapidly switch beam between broadside patterns for wide coverage and end-fire patterns for directional links, while reflector boards enhance gain and reduce interference [128].

#### 4.3.3. Mechanically Steerable Platforms

Mechanically steerable platforms use motor-driven rotation or tilt to physically direct antenna beams, making them effective for vessels constantly in motion [129]. Unlike complex electronic beamforming, they ensure reliable line-of-sight connectivity and can support high-gain antennas, which makes them well-suited for larger ships. Recent studies [130,131] highlight the importance of these techniques in dynamic environments, particularly for underwater communications and the development of next-generation adaptive networks.

#### 4.3.4. Hybrid Mechanical–Electronic Beam Steering

Hybrid mechanical–electronic beam steering combines fast electronic adjustments over a limited angular span with slower mechanical movements to achieve broader coverage [132]. This approach allows antennas to track satellites or vessels across wide azimuth ranges while lowering both mechanical complexity and power consumption. It further improves system robustness, enables flexible beam shaping, and helps maintain stable link quality in dynamic maritime environments [133]. A practical example is found in recent studies on hybrid multibeam receiver designs, which confirm its suitability for LEO satellite communications at Ku-band [103].

### 4.4. Energy Efficiency in Maritime Adaptive Antennas

Energy efficiency is a constant problem for maritime IoT devices since most depend on batteries or small solar panels that provide only limited power [134]. Adaptive antennas help by adjusting transmit power as conditions change with the channel or vessel movement. The objective is to maintain reliable connectivity while minimizing energy expenditure by avoiding redundant transmissions [135,136,137].

Another way to save power is through sleep–wake scheduling. Instead of keeping nodes active all the time, activity can be timed to match vessel operations or periods when traffic is expected. This simple step sharply cuts energy use while maintaining communication reliability in the shifting conditions at sea [138,139].

### 4.5. Maritime Channel Modeling

Radio channels at sea behave very differently from those on land, which makes modeling essential [140]. Signals often arrive along two paths, one direct, the other reflected from the water surface. The two-ray model is commonly used to describe this, providing estimates of power variation that feed directly into antenna design [141]. Conditions worsen when the sea is rough. Vessel motion and wave scattering break the simple two-path picture, producing fading that is usually treated with the Rayleigh roughness parameter [142,143]. Close to the coast, propagation is influenced by additional factors: atmospheric layering, surface refractivity, and even the curvature of the Earth. Models such as AREPS are used in these cases to account for ducting and unusual refraction [144,145]. Weather then adds another complication. Rain, fog, and snow all absorb energy, and the effect is strongest at higher frequencies. The Crane model is often applied here to predict losses, while practical countermeasures such as adaptive equalization and SNR tracking are needed to keep the link stable [146,147].

### 4.6. Maritime Interference Mitigation

Maintaining stable communication at sea is complicated by spectrum congestion and elevated noise, both of which contribute to persistent interference and reduced link performance. The adaptive null steering technique works by continuously adjusting the antenna’s radiation pattern to suppress interference and improve link quality. This technique has been shown to improve performance even when hardware or propagation conditions are not ideal [148,149]. Adaptive antennas are also being applied to navigation systems. In the case of AIS, for example, they extend range and improve throughput. At the same time, security enhancements are achieved by pairing with the VHF Data Exchange System (VDES) for higher-rate links and with Public Key Infrastructure (PKI) for authentication of messages [150,151,152,153]. The VHF marine band poses its own challenges, and interference there is often handled with adaptive notch filtering together with real-time spectrum monitoring. These methods cancel narrowband interferers and help avoid occupied voice channels, which keeps links from being disrupted [154,155]. Satellite communication is no less demanding. Side-lobe and adjacent-terminal interference are suppressed by phased arrays, beamforming, and polarization filtering, making it possible for ship terminals to operate alongside powerful offshore satellite uplinks [13,156].

## 5. Adaptive Antenna and LoRaWAN

### 5.1. Adaptive Antennas at Gateway and End-Node Levels

Adaptive antennas improve LoRaWAN performance at sea, but they do so in different ways at gateways and at end nodes. Gateways have the resources to run complex schemes, such as phased arrays, switched beams, or even multi-beam receivers, steer signals, raise SNR, and reduce packet loss [104,157]. Trials such as MoLoRa show that reorienting antennas can overcome blind spots and multipath [158]. At the same time, other studies report longer detection ranges and stronger links using Non-Uniform Linear Array (NULA) beamforming and Minimum Variance Distortion-less Response (MVDR) filtering [159]. More recent work adds refinements like convex-optimization-based shaping [155] and dual-feed phased arrays with polarization control [156,157]. Together, these show the kinds of robustness gateways can afford.

End nodes tell a different story. Buoys, sensors, and small vessels run on tight power budgets, so energy efficiency takes priority [160]. Here, simple switched-beam or reconfigurable designs make sense, sending directional signals only when needed. This saves energy without losing reliability. Yet, practical issues, such as waterproofing, corrosion, and the instability of floating platforms, still limit deployment. Encouragingly, compact polarization-reconfigurable antennas already demonstrate low-power adaptability [161], and a 915 MHz digital phase-shifter antenna has proven that efficient beam steering can be achieved in small IoT devices, making it highly suitable for maritime LoRaWAN nodes where focused signals and interference rejection are crucial [105].

### 5.2. Medium Access Control (MAC) and Physical Layer (PHY) Considerations

In maritime LoRaWAN networks, adaptive antennas contribute at two levels; they improve link quality at the physical layer and help maintain efficiency and reliability at the MAC layer. Techniques such as beam steering and directional transmission can lift the SNR considerably, but they only work well when the system can track the channel in real time and use feedback to keep the beams aligned [162]. This real-time CSI may involve some protocol support or hardware changes for the LoRa transceiver so that it supports antenna switching or reconfiguration. LoRaWAN’s standard pure Additive Links On-line Hawaii Area (ALOHA) protocol operates at the MAC layer in an uncoordinated manner, with no provisions for spatial directionality control or adaptive antenna state transition [163]. Alternatively, suppose nodes employ adaptive antenna behavior to avoid collisions or minimize packet loss by transmitting/broadcasting only when no transmission is taking place in the broadcast environment, and the node is allowed to transmit. In that case, policy may accidentally cause collisions [164]. To address this problem, improved MAC schemes, time-slotted access, priority-based scheduling, or group-based directional contention management are being investigated to coordinate better transmission timing with antenna configurations.

When gateways use directional antennas, precise synchronization with devices is critical to avoid uplink reception issues in dynamic maritime setups [165]. Cross-layer approaches that combine physical-layer conditions with MAC scheduling and beam-state selection have been shown to improve throughput and overall network performance under resource-constrained maritime conditions [166]. Figure 8 shows the LoRaWAN protocol stack by which the MAC layer (supporting Class A, B and C modes) and LoRa modulation run on regional ISM radio bands under the application layer.

### 5.3. LoRaWAN Parameter Optimization

Integrating adaptive antennas into maritime LoRaWAN requires rethinking key transmission parameters, since beamforming and reconfigurability dynamically alter the wireless channel. Parameters such as Spreading Factor (SF), Transmission Power (Tx), Bandwidth (BW), and Coding Rate (CR) directly affect energy efficiency, reliability, and the performance of the Adaptive Data Rate (ADR) mechanism, which balances throughput, stability, and power consumption [66,168].

#### 5.3.1. Spreading Factor

A key characteristic of LoRaWAN is the balance it strikes between range and data rate, controlled by its configurable SF settings. Lower SFs enable higher data rates but reduce coverage, whereas higher SFs extend the range but come with increased latency and reduced throughput [23,169]. SF affects both the Time-on-Air (ToA) and the communication range [170,171]. ToA defines the transmission duration of a LoRaWAN packet and depends on the chosen SF, BW, and CR [14]. By improving SNR, adaptive antennas allow devices to use lower SFs, which reduces delay and energy consumption. De Jesus et al. [172] put forward a variable-margin ADR scheme that tunes the SF to the link quality, and they noted it could be even more effective alongside adaptive antennas.

#### 5.3.2. Transmission Power

LoRa devices typically operate at 13–20 dBm depending on the frequency band [173]. Tx settings are constrained by duty-cycle regulations (ranging from 0.1% to 10% in the EU868 band), as well as regional rules on ToA, channel hopping, and packet duration [174]. Tx power optimization, when combined with adaptive antennas, enhances link robustness while maintaining fair spectrum use.

#### 5.3.3. Bandwidth

LoRaWAN operates in unlicensed bands (433, 868, 915 MHz), with default BW at 125 kHz but extendable up to 500 kHz [35]. Wider BWs allow faster transmission but can raise energy consumption and reduce signal integrity. Standardized widths of 125, 250, and 500 kHz are typically used [175], and adaptive selection becomes important for balancing efficiency and robustness in dynamic maritime channels.

#### 5.3.4. Coding Rate

CR controls error correction and efficiency. In LoRaWAN, CRs of 4/5, 4/6, 4/7, and 4/8 describe the balance between bits used for forward error correction and those used to carry the actual payload [176]. Lower CRs make communication more reliable but less efficient in noisy maritime conditions, while higher CRs allow faster data transfer but increase the chance of packet loss. The ideal CR is application-specific and is usually set through field tests and simulations [23,112].

#### 5.3.5. Adaptive Data Rate

ADR dynamically adjusts SF, Tx power, and CR to conserve energy while keeping the network efficient. These changes are guided in real time by link quality indicators such as SNR, RSSI, and packet success rate [177,178]. For fixed devices, network-managed ADR works well enough, but mobile nodes often need blind ADR, which lets them adapt without feedback from the gateway [176]. The drawback is that both approaches break down when the channel changes too quickly, as it often does at sea. One way around this is to pair ADR with adaptive antennas and adaptive modulation [179].

### 5.4. Challenges in Integration

Bringing adaptive antennas into LoRaWAN at sea is not straightforward. The difficulties come from several directions: the lack of standardization, the need for close hardware–software integration, and the unpredictable nature of the marine environment. The LoRaWAN specification itself sets out frequency bands, data rates, and device classes, but it does not provide for adaptive features such as beamforming or dynamic pattern control. This gap creates a risk of non-compliance, especially with strict limits on antenna gain and radiation patterns [180]. On the implementation side, adaptive systems require seamless integration of hardware components (antenna arrays, RF transceivers, microcontrollers) with software tasks such as beamforming and control, which is difficult in practice because it demands real-time processing, fast inter-module communication, and tight power management in constrained devices [181]. These challenges are further intensified by the harsh nature of the maritime environment, where sea-surface reflections, vessel movement, and irregular propagation channels give rise to multipath fading and interference. As a result, antennas mounted on buoys and ships must constantly adapt to preserve reliable communication links [35].

Integration aspects at both the gateway and end-node levels, including feasibility, challenges, and opportunities for maritime IoT, are outlined in Table 5.

## 6. Results and Discussion

An organized description of the five RQs of this systematic review is provided in this section. For each question, the synthesis of results from the 23 selected articles is applied, focusing on common threads, deployment schemes, and open issues of applying adaptive antenna systems in maritime LoRaWAN.

### 6.1. Thematic Analysis

The dominant research themes of adaptive antenna for maritime LoRaWAN applications are visualized in Figure 9, which has been constructed by the technical terms that have been selected from the literature review. The visualization highlights key technology principles incorporating adaptive antenna, beamforming, phased array, and smart antenna, and regards the increasing presence of intelligent spatial filtering solutions in LPWAN. Words such as multipath, interference, and radiation pattern emphasize the major communication bounds in the maritime environment. In contrast, SNR, RSSI, Doppler shift, and fading describe the average performance parameters that are commonly employed to evaluate link reliability in sea state-affected channels. The SF, the MAC layer, the duty cycle, and the gateway that were included in this review also demonstrate architectural and protocol aspects of LoRaWAN optimization. Altogether, the keyword distribution reveals a very technical interest in physical layer improvements as well as network-level improvements to accommodate the specific requirements of long-range and energy-efficient communication in the marine context.

### 6.2. RQ1: What Are the Adaptive Antenna Techniques That Contribute Most to the Energy Efficiency and Performance on Maritime LoRaWAN?

The review categorizes four types of adaptive antennas for maritime LoRaWAN. Switched-beam antennas are based on a fixed set of radiation patterns. They are of small size, therefore appropriate for low power needs (e.g., buoys with small sea sensors), which gives a quick directionality and energy-efficient beamforming at low computational complexity [79,84]. Phased array antennas electronically steer their beams by adjusting the phase across multiple radiating elements, which allows dynamic link tracking while providing high gain and a narrow beamwidth. These are well-suited to gateway-level deployments, such as for coastal base stations or ship relays [86,104,105]. Reconfigurable antennas that vary the frequency, radiation pattern, or polarization using MEMS or varactors demonstrate frequency agility and are adaptive to changing interference, and are therefore well suited for mobile maritime platforms [87,88,89,103]. Lastly, the AI/ML-assisted smart antennas can dynamically adjust their beam patterns and the operation parameters in real time while learning, providing advanced adaptivity for smart marine scenarios with diverse mobility, interferences, and varying environments [78,92,182]. Figure 10 represents the estimated distribution of the types of adaptive antennae introduced in this review for maritime LoRaWAN networks.

Based on the literature reviewed, four main types of adaptive antennas are employed in the maritime LoRaWAN systems. Switched-beam antennas dominate at 45%, largely because their simple, low-cost design fits well with power-constrained platforms such as marine sensors and buoys. Phased array antennas follow at 30%, as they enable high gain along with dynamic beam steering and are frequently used for infrastructure-level applications such as coastal gateways or ship relays. Reconfigurable antennas form 20%, allowing them to deal with environmental variation, although for mobile marine platforms, the ability to adapt the pattern or the frequencies in real time could be considered. AI/ML-powered antennas, which are in their early stages, constitute 5% of use cases; capabilities are advanced adaptability and learning-based optimization for intelligent maritime systems.

### 6.3. RQ2: What Are the Effects of Marine Environmental Conditions on Adaptive Antenna Response?

In the maritime setting, wireless links are rarely stable because vessels and buoys are constantly moving. Pitch, roll, yaw, wave height, and even weather shifts can throw an antenna off balance and affect its performance [41]. On moving platforms such as buoys or autonomous surface craft, the problem is worse, since a small change in motion can alter the elevation angle and break beam alignment [31]. To cope with it, elevation-compensating beamforming and real-time mechanical stabilization with the help of motion sensors have been used to maintain link continuity during navigation in rough seas [94]. Sea surface reflections also generate strong multipath reflections that decrease the signal quality, particularly at low antenna elevations [62]. Adaptive beam forming schemes, e.g., horizon-focused and multipath adaptive steering, have been proposed to mitigate such effects by concentrating the energy on the optimum horizontal path and suppressing late signals [104].

In addition, meteorological conditions such as humidity, fog, and rain are responsible for signal attenuation and must be considered in antennas and housings. It is typical of maritime antennas to incorporate environmental protection technologies, in particular IP68-rated enclosures and corrosion-resistant coatings, to exert robustness against severe environmental conditions. As a result, maritime-tailored adaptive antennas are increasingly designed with environmental awareness, thus making the LoRaWAN reliable communication despite sea-state variations, atmospheric interference and vessel mobility [71,157].

### 6.4. RQ3: What Are the Key Antenna Parameters That Affect Reliable Maritime LoRaWAN Communication?

Several performance parameters emerge as critical for reliable maritime LoRaWAN links: gain, beamwidth, directivity, reconfigurability, and polarization agility. The author in [84] showed that a switched-beam antenna improved coverage through beamwidth optimization with minimal energy use, while [87,105] showed how gain and directivity support stable long-range links. Reconfigurable designs in [88,89] adjusted frequency or pattern, boosting energy efficiency without reducing performance. Polarization control, such as in phased arrays [104], further reduced interference from sea-surface reflections. The triple-band monopole in [103] showed that multiband capability could extend application versatility while maintaining range. These parameters determine how well an antenna can maintain SNR and packet delivery under maritime constraints.

Figure 11 shows that three parameters matter most for maritime LoRaWAN antennas: beamwidth, gain, and energy efficiency. Narrow beams help cut interference. Higher gain stretches the coverage range. Efficiency matters because it lets nodes run at lower power levels. The role of other features is less clear. Polarization control, reconfigurable antennas, and AI/ML adaptivity show promise for improving robustness, though the findings are inconsistent across different maritime conditions. The added hardware and computational load make them difficult to apply in small, energy-constrained devices.

### 6.5. RQ4: Which Antenna Designs Are the Best Interns in Terms of Performance, Energy Efficiency, and Complexity?

To study the performance, power consumption, and system complexity trade-offs, different implementation strategies have been investigated at both gateway and end-node levels. Gateway-based systems often employ phased arrays or switched-beam architectures, which allow precise beam control, deliver significant SNR improvements, and enable multiple connections at once. Although advanced techniques can achieve notable performance gains, they introduce trade-offs. Systems that depend on multiple RF chains and high-speed signal processors tend to be more complex and significantly more costly to implement [86,93]. For that reason, much of the focus at the end-node level has shifted toward simpler, low-power solutions. Reconfigurable and polarization-switchable antennas are good examples, as they can be tuned with only lightweight control, keeping the design efficient and practical for real deployments [88,89].

Apart from those widely used configurations, some other practical realizations have also been investigated to accommodate deployment-dependent limitations. Diode-controlled parasitic array antennas offer a low-complexity, compact alternative for smart buoys and mobile sensors [92,135]. Mechanically steerable antennas are slower to respond but have solid performance on vessel-mounted platforms where orientation is changed using tilt and rotation mechanisms [79,94]. Such hybrid systems, capturing the benefits of both electronic and mechanical steering, represent evolving solutions to the compromise between wide-angle coverage and limited power generation in dynamic but highly variable sea-states and high-mobility environments [26,103].

A summary of the main characteristics and trade-offs of the adaptive antenna systems of the review is shown in Table 6, with each of these being mapped to their deployment levels, SNR gain estimations, energy consumption, and environmental compatibility.

Earlier systems, such as Multi-Antenna LoRa (MALoRa) and Mobile LoRa (MoLoRa), provide the basis for the SNR gains summarized in Table 6 [100,158]. MALoRa reported improvements of up to 10 dB through multi-antenna signal combining under controlled test conditions. In comparison, MoLoRa achieved as much as 13 dB using mobile antenna adaptation in dense urban trials. Neither of these studies wer carried out in a maritime setting, but together they provide useful lower- and upper-bound references. More importantly, they point to the potential of adaptive beamforming and diversity methods to produce meaningful improvements in LoRaWAN. These results give a realistic sense of what may be achievable when the same ideas are applied under the far harsher conditions at sea [183].

### 6.6. RQ5: What Are the Research Gaps and Future Directions?

Table 7 summarizes the main challenges for integrating adaptive antennas with maritime LoRaWAN, including issues with standardization, hardware–software co-design, environmental modeling, and access to open datasets. RIS is a promising avenue, together with lightweight AI and cross-disciplinary collaboration, for driving the next stage of research in this field.

### 6.7. Synthesis of the Findings

This review looked at how adaptive antennas can support maritime LoRaWAN, particularly in terms of coverage, reliability, energy efficiency, and resilience to harsh conditions. Since the studies used a wide range of methods and setups, it was not possible to perform a formal meta-analysis. Even so, a clear trend appeared. Adaptive antennas improved performance almost across the board, with reported gains in signal-to-noise ratio, packet delivery ratio, and reductions in time-on-air [85,86,143,144,146,152]. For small, low-power devices such as buoys, switched-beam and reconfigurable designs emerged as the most practical. Phased arrays, on the other hand, were better suited to gateways, where power and processing are not as restrictive [85,86,143,152]. While the overall evidence is encouraging, the absence of long-term sea trials remains a serious gap. Without them, these findings remain largely confined to controlled or short-term studies.

Some of the limitations are structural. LoRaWAN protocols still lack native support for adaptive antenna functions, so the MAC and ADR layers cannot yet exploit directional communication fully [22,31]. Hardware–software integration is another sticking point, especially at the end node, where synchronizing RF switches, beamforming circuits, and control logic is constrained by tight energy budgets [36,49,103]. Environmental modeling lags even further behind. Existing models account for only part of the complexity caused by varying sea states, vessel motion, and multipath fading, which reduces the accuracy of both prediction and adaptation [35,41,46,107]. The absence of open datasets and standardized testbeds further complicates progress, as it restricts reproducibility and fair comparison across studies [104,105]. The lack of open datasets and standardized testbeds compounds the problem, making reproducibility difficult. Compact and modular antenna designs for small marine sensors are also rarely studied, even though they could provide a practical path to scalable deployments.

Future directions are clear enough. Lightweight AI models that can adjust beam direction, tilt, and transmission power with minimal energy cost will be needed if adaptive approaches are to be viable at scale. Reconfigurable intelligent surfaces add another promising layer, with potential to improve spectral efficiency and extend coverage under complex maritime conditions. A more fundamental issue is cross-medium propagation; the air–sea and underwater boundaries remain poorly understood and need dedicated study. Above all, reproducibility requires standardized long-term testbeds and shared datasets, without which results cannot be fairly compared. Furthermore, real progress will depend on closer collaboration between wireless engineers, ocean scientists, and system designers. Only with this kind of interdisciplinary work can maritime IoT infrastructures become both scalable and resilient.

## 7. Conclusions

Adaptive antennas show strong potential for improving LoRaWAN performance at sea. The most suitable designs depend on the device; switched-beam and reconfigurable antennas are better for small, battery-powered nodes, while phased arrays and beamforming are more practical at gateways with higher power and processing capacity.

Despite these advances, several barriers remain. LoRaWAN standards do not yet support adaptive antenna functions. Hardware–software integration is still difficult in energy-limited devices, and present models fail to capture the fast-changing nature of the sea. The lack of shared testbeds also makes it hard to validate results over time.

Progress will require lightweight beam control methods, exploration of RIS, lower frequency bands such as 433 MHz and long-term trials in real maritime conditions to enhance the robustness of maritime LoRaWAN deployments further. Just as important, collaboration between communication engineers, ocean scientists, and system designers will be needed to turn these experimental gains into reliable practice.

## Figures and Tables

**Figure 1 sensors-25-06110-f001:**
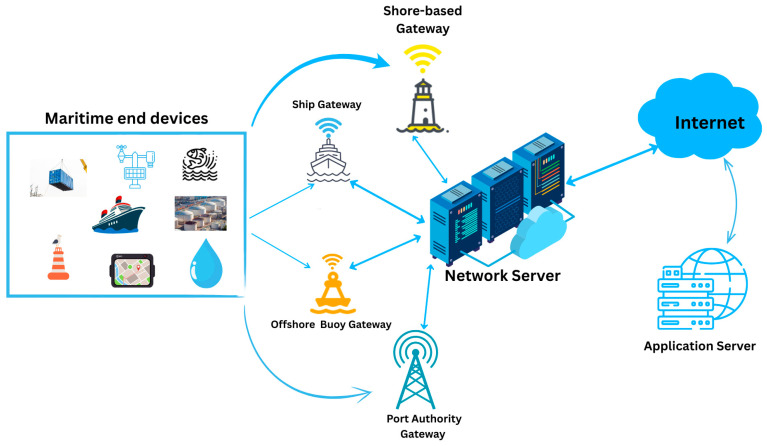
Maritime LoRaWAN architecture.

**Figure 2 sensors-25-06110-f002:**
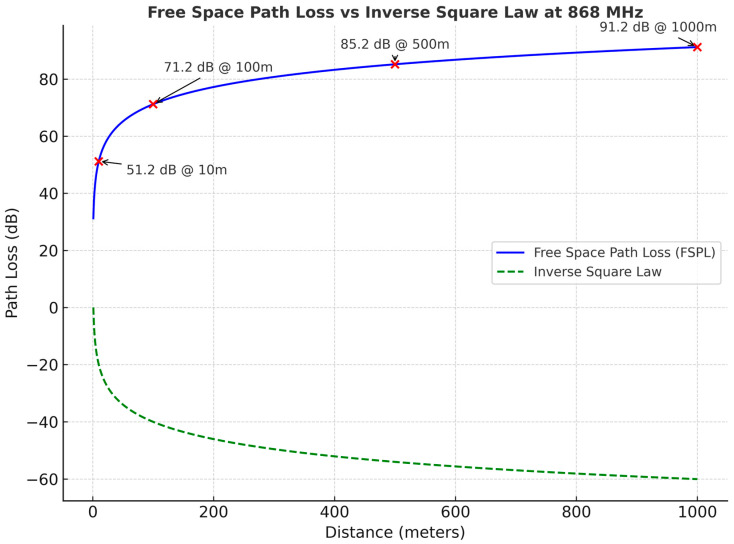
FSPL model.

**Figure 3 sensors-25-06110-f003:**
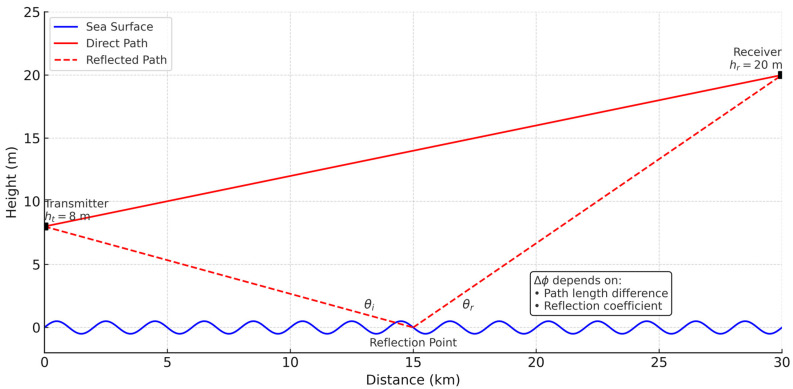
Maritime two-ray propagation model.

**Figure 4 sensors-25-06110-f004:**
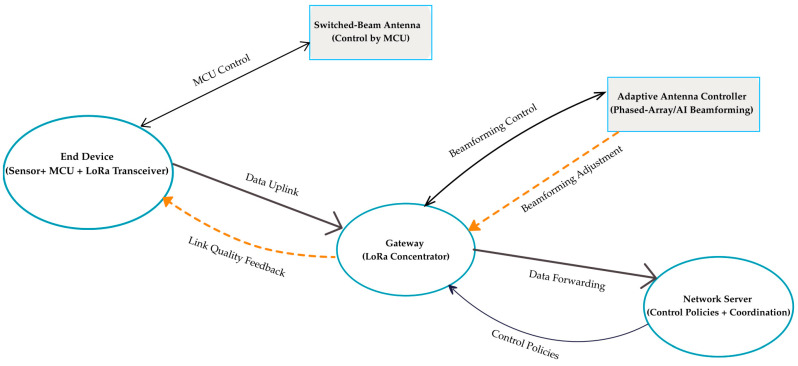
Control and integration of adaptive antennas in LoRaWAN.

**Figure 5 sensors-25-06110-f005:**
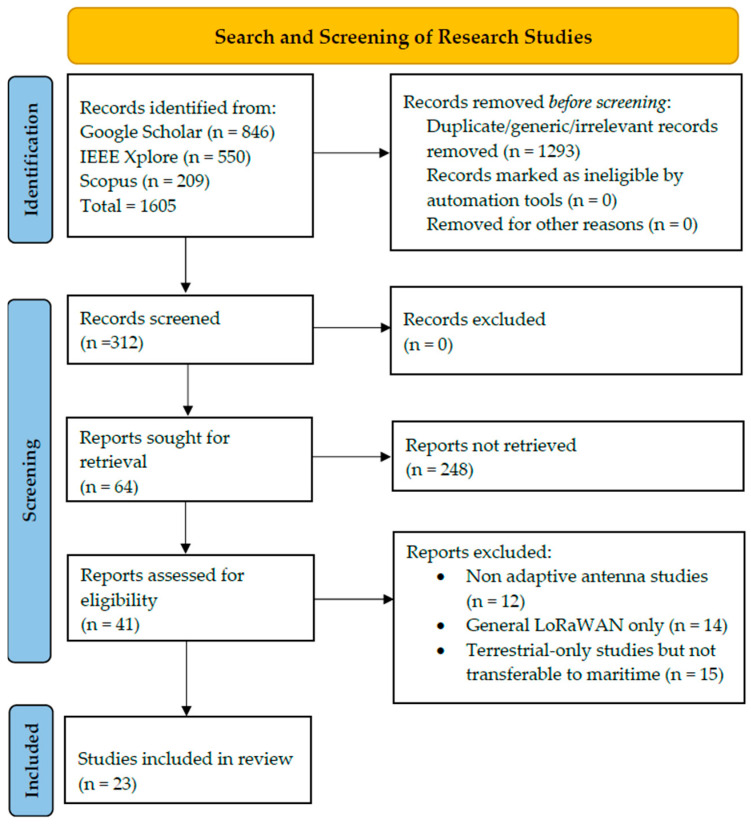
Systematic literature review flow diagram.

**Figure 6 sensors-25-06110-f006:**
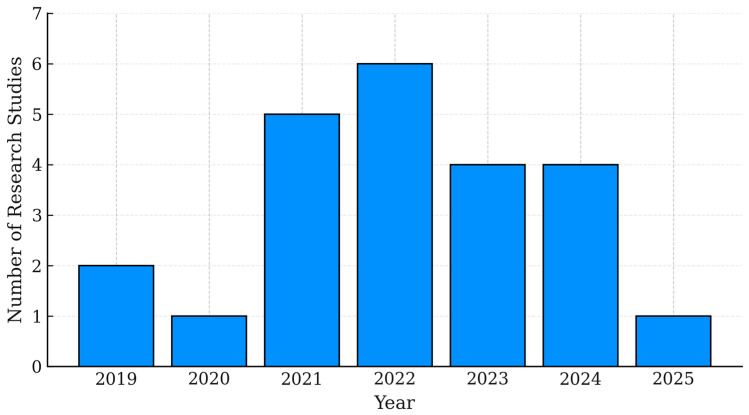
Number of selected research studies per year (2019–May 2025).

**Figure 7 sensors-25-06110-f007:**
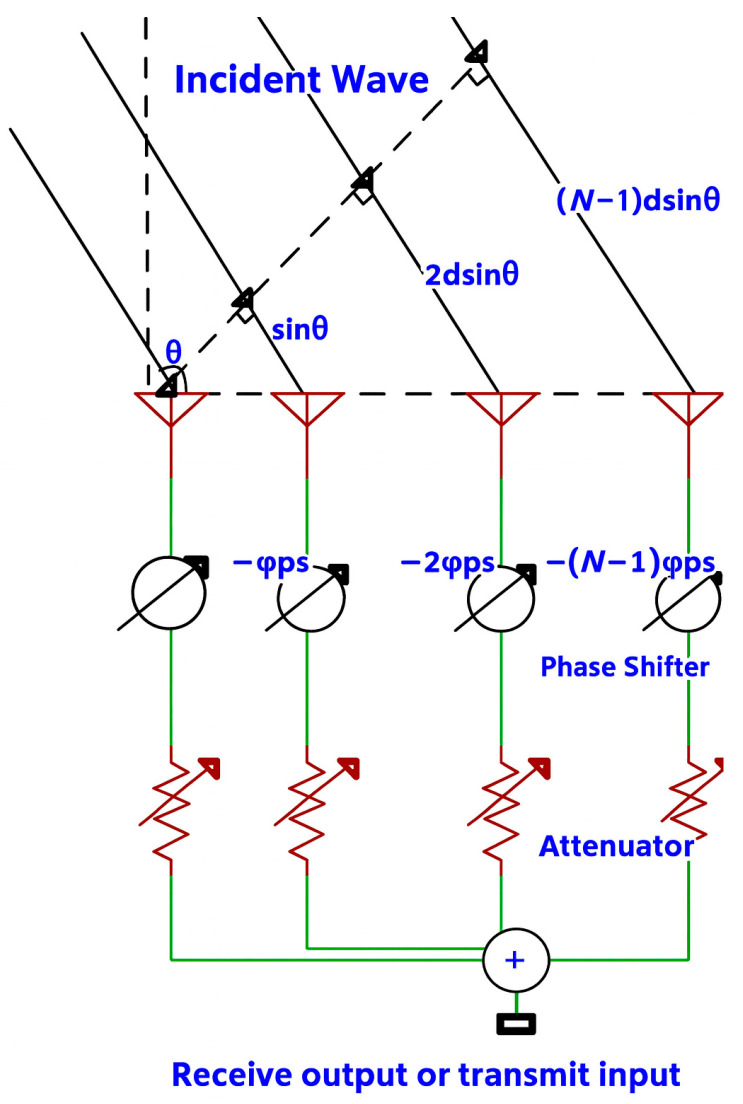
PAS transmit or receive [109].

**Figure 8 sensors-25-06110-f008:**
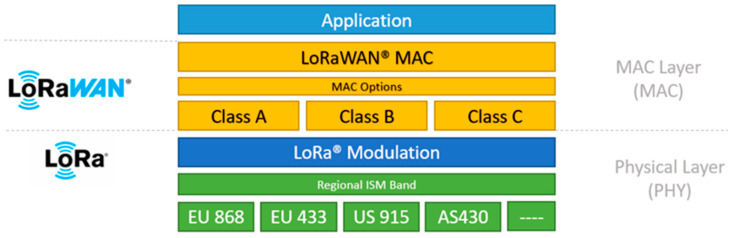
LoRa protocol stack: PHY and MAC layers, adapted from [167].

**Figure 9 sensors-25-06110-f009:**
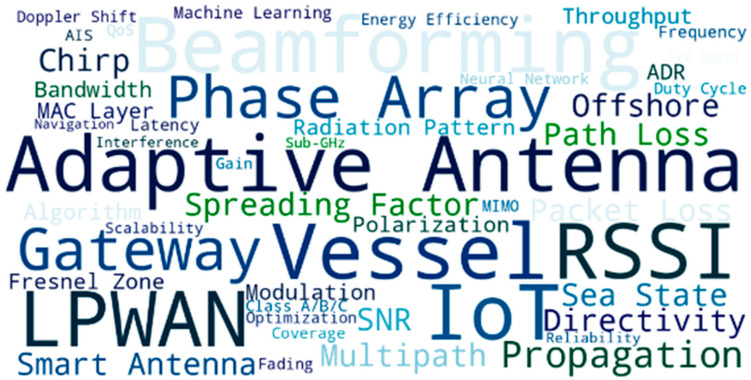
Summary of the most frequently occurring technical terms in this review.

**Figure 10 sensors-25-06110-f010:**
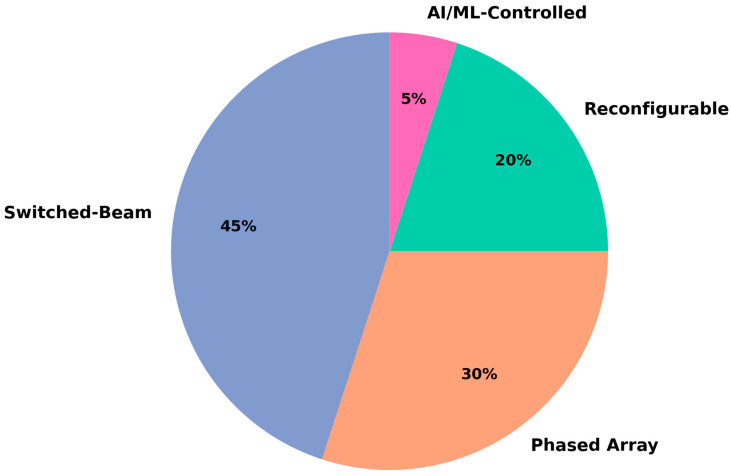
Estimated distribution of adaptive antenna types identified in this review.

**Figure 11 sensors-25-06110-f011:**
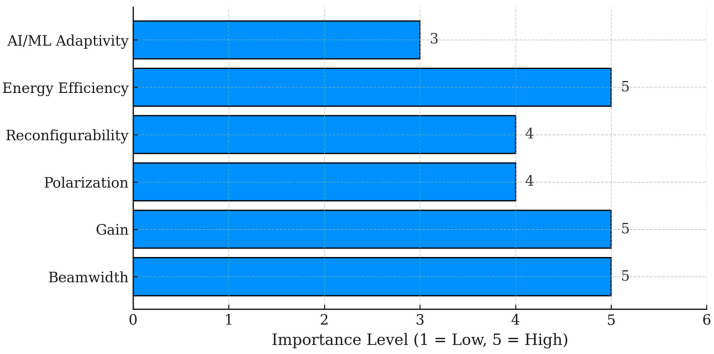
Key antenna parameters in LoRaWAN.

**Table 1 sensors-25-06110-t001:** Wireless technologies appropriate for marine conditions.

Article	Technology	Frequency Band	Range (About)	Data Rate	Benefits of Marine Applications	Restrictions in Marine Applications
[19,50]	LoRaWAN	868 MHz in Europe and 915 MHz in the USA	50 km	0.3 to 50 kbps	Low power, long range, and appropriate for remote monitoring (vessels, buoys, and sensors)	Data rate limitations and interference from marine obstacles
[51]	Narrowband Internet of Things (NB-IoT)	Long-Term Evolution (LTE) Bands with Licenses (1800 MHz and 2100 MHz)	35 km	250 kbps	Reliable connectivity to LTE networks, suitable for offshore asset monitoring	Needs network coverage and potentially higher latency
[52]	Zigbee	2.4 GHz	100 m	20 to 250 kbps	Low power consumption is helpful for onboard vessel networks	Short range and impacted by interference in marine settings
[53]	Wireless Fidelity (Wi-Fi) 802.11ah	Sub-GHz (750–950 MHz)	Several km (favorable conditions)	Up to ~78 Mbps	High data rates are beneficial for shipboard communications	High power usage and a short range over the water’s surface
[54]	Bluetooth Low Energy (BLE)	2.4 GHz	100 m	2 Mbps	Ideal for short-range Internet of Things applications on ships	Interference and short range in maritime settings
[55]	Fifth Generation (5G)-SA/Private	Different (Sub-6 GHz, mm Wave)	10 km (macro >10 km)	Maximum 10 Gbps	Analysis in real-time and high-speed data transfer for smart ships	Limited coverage offshore and high-power consumption
[56]	Satellite Internet of Things (IoT)	GHz Bands	Global	Differs	Designed for deep-sea communication and emergency connectivity	Expensive, slower and weather-dependent
[57]	High Frequency (HF) Radio	3–30 MHz	100s–1000s km	Low	Long-distance communication between ships and the shore	impacted by atmospheric circumstances, low data rate
[57]	Very High Frequency (VHF) Radio	30–300 MHz	100 km	Low	For communication between ships and between ships and shore	Restricted bandwidth and license requirements
[58]	Automatic Identification System (AIS)	161.975 MHz & 162.025 MHz	75 km	9.6 kbps	Crucial for tracking vessels and preventing collisions	Not for general communication, only for maritime navigation
[59]	LTE-M	Licensed LTE bands	1–11 km	≤1 Mbps	Supports mobility, voice, and low-latency IoT	Requires LTE infra, higher power than LoRa
[60]	Acoustic/underwater	~10 kHz to a few MHz	10 km	Few kbps	Vital for AUVs and underwater sensor networks	High latency and restricted data rates

Units: kHz = kilohertz, MHz = megahertz, GHz = gigahertz, m = meter, km = kilometer, bps = bits per second, kbps = kilobits per second, Mbps = megabits per second, Gbps = gigabits per second, mm Wave = millimeter wave, dB/km = decibels per kilometer.

**Table 2 sensors-25-06110-t002:** Summary of maritime propagation models.

Article	Model	Key Feature	Application	Limitations
[76]	FSPL	Ideal LoS attenuation	Benchmark and baseline coverage	Does not take into account reflections, diffraction and atmospheric effects.
[62]	Two-ray ground	Reflective sea surface interference	Calm sea, low-height antenna deployments	Fails in rough seas; neglects atmospheric refraction
[72]	ITU-R P.1546	Practical long-range signal prediction	Regulatory considerations, wide-area LoRaWAN networks	Needs local calibration; conservative in variable seas
[73]	Knife-edge diffraction	Bending of the signal around an obstacle with sharp edges	Nearshore facilities and vessels	Models exactly one ideal obstacle
[74]	Tropospheric ducting	Long-range extension driven by the weather	Auxiliary considerations for expanded coverage	Unforeseen, based on passing circumstances
[75]	Empirical measurement	Field-based, site-specific modelling	Fjords or offshore site small-scale optimization	Not portable: needs a measurement campaign

**Table 3 sensors-25-06110-t003:** Trade-offs between adaptive antenna types.

Article	Antenna Type	Performance	Complexity	Power Consumption	Compatibility For Marine LoRaWAN
[81,84]	Switched-beam	Reasonable gain, unidirectional coverage	Low (simple switching logic)	Low	Good for low-cost power-limited setups
[86]	Phased array	High gain, fast beam steering	High (phase control circuits are necessary)	Medium to high	Perfect for high-performance or dynamic maritime links
[89]	Reconfigurable	Adaptable pattern, frequency agility	Medium (control circuits are required)	Medium	Flexible choice for active marine environments
[92]	Smart AI/ML-controlled	Adaptive, self-learning pattern optimization	Quite high (needs AI/ML models)	Medium to High	Ideal for autonomous or intelligent ships with real-time channel adaptation
[101]	MIMO	High-capacity, multipath robustness	Very high	High	Applicable to shoreside or shipside gateway
[35]	Omnidirectional	Uniform coverage, no beamforming	Very low	Very low	Best for simple nodes with minimal complexity

**Table 4 sensors-25-06110-t004:** Summary of the included studies.

Article	Year	Focus	Key Findings
[26]	2022	Energy	Multi-layered energy efficiency strategies involving adaptive antenna control.
[31]	2019	Performance	Offshore aquaculture data are transmitted by LoRaWAN with a range of 8.33 km (LoS), achieving a maximum of 87.33% packet reception at optimal SF7–10 with limited Fresnel clearance and harsh sea conditions.
[35]	2021	Performance and energy	LoRaWAN coverage extended up to 40 km offshore; environmental factors strongly influenced link reliability.
[41]	2021	Environment	Effects of tropical weather on LoRaWAN communications.
[71]	2022	Performance	Validated LoRaWAN channel models for complex estuarine environments.
[84]	2019	Performance	A small switched-beam antenna for 868 MHz IoT applications with beam steering capability to improve the range and reliability in low-power deployments is introduced.
[87]	2024	Performance	Outlines microstrip designs for maritime adaptation.
[88]	2023	Energy and performance	The study proves that reconfigurable antenna design enables higher coverage and energy efficiency.
[89]	2024	Performance and energy	Highlights dynamic tuning for LoRaWAN efficiency.
[81]	2021	Energy and Performance	Implemented a switched-beam array to enhance link margin and energy efficiency in mobile LoRa nodes.
[100]	2022	Performance	Demonstrated that multi-antenna gateways using MIMO and beamforming can nearly double network throughput.
[103]	2025	Performance	Triple-band reconfigurable monopole antenna adapted for maritime LoRaWAN.
[46]	2022	Energy and Environment	Demonstrated feasibility of bridging underwater acoustic sensor networks with above-water LoRaWAN gateways for marine IoT.
[104]	2022	Performance	Improvement of maritime link quality using beamforming.
[105]	2023	Performance	Demonstrates an array design applicable to long-range LoRaWAN.
[36]	2024	Performance and Environment	A multi-gateway LoRaWAN system tracked a boat over 16 km with gateway ranges up to 5.7 km.
[106]	2021	Performance	Novel LoRa antenna with oil-paper buffer achieved 6 m underwater and 160 m surface range.
[107]	2024	Energy and Environment	In Sardinia, solar-powered LoRaWAN buoys were used to keep track of marine weather and water quality, showing that sustainable energy can support continuous monitoring.
[108]	2023	Performance and Environment	A system combining LoRaWAN with BLE and GPS made it possible to follow boat movements in real time, even in shallow waters where interference is common.
[109]	2023	Environment and performance	By using multi-hop LoRa, researchers managed to push coverage beyond the usual single-hop range, which improved both the reliability and the overall data throughput.
[110]	2021	Performance	Developed a compact PIN-diode reconfigurable antenna switching between UHF and LoRa bands.
[111]	2020	Performance	Demonstrated beam-steering antenna array at 868 MHz tailored for LoRa/LPWAN gateways.
[112]	2022	Performance	Frequency optimization for LoRaWAN using adaptive antenna tuning in maritime contexts.

**Table 5 sensors-25-06110-t005:** Summary of antenna integration in LoRaWAN.

Article	Aspect	Gateway Level	End Node Level
[104,157]	Integration level	Centralized, supports multi-beam and multi-node handling	Distributed, constrained by energy and form factor, static, low power
[104,157,160,161]	Antenna technologies	Phased Array, switched-Beam, multi-Beam antennas	Switched-beam, reconfigurable compact antennas
[88,160]	Power constraints	Ample power from grid/solar/diesel sources	Battery or small solar, limited-duty cycles
[92,161,182]	Computational requirements	Supports DSP, AI models, and real-time beam steering	Low-power MCUs, reconfiguration triggered by events
[104,157]	Beamforming features	Supports dynamic steering and simultaneous beam formation	Supports basic beam direction switching, limited steering
[78,79,84,111]	Signal quality impact	High SNR, reduced packet loss, supports mobile nodes	Moderate SNR gain, challenged by node drift/misalignment
[87,88,94,161]	Environmental adaptability	Suited for fixed installations or large mobile platforms (e.g., ships)	Needs waterproofing, orientation control, and compact design
[94,158,163,164]	MAC layer implications	Directional MAC extensions are needed to prevent beam conflicts	Standard MAC lacks support for directional scheduling
[78,86,158,162]	PHY layer enhancements	Affects the link budget, requires CSI for beam alignment	Limited influence, needs PHY-level switch integration
[112,172]	Parameter optimization	Real-time SF, TX, CR adjustment with adaptive feedback	gateway-assisted ADR optimization
[78,182]	AI/ML capability	Supports inference-based beam selection and link prediction	Limited only to lightweight or gateway-driven models
[88,104,161]	Deployment suitability	Coastal gateways, ship relays, offshore stations	Smart buoys, marine sensors, mobile aquatic UAVs
[88,165,181]	Integration challenges	Protocol upgrades, real-time sync, and MAC–antenna coupling	Hardware–software co-design, waterproofing, energy balance

**Table 6 sensors-25-06110-t006:** Summary of main characteristics and trade-offs of adaptive antenna systems.

Article	Antenna Type	Deployment Level	SNR Gain (dB)	Energy Cost	Maritime Suitability	Use Case
[86,105,182]	Phased array	Gateway	8–10	High	High (for fixed installations)	Shoreline gateways, ship relays
[84,104,157]	Switched beam	Gateway/End-node	4–8	Moderate	Medium (variable performance on mobile platforms)	Ship antennas, smart buoys
[87,88,89]	Reconfigurable Patch	End-node	3–5	Low	High (compact, suitable for mobile nodes)	Drifting sensors, remote buoys
[87,89]	Polarization-reconfigurable	End-node	2–4	Very Low	High (low power and orientation-tolerant)	Wearable or onboard devices
[93,94]	Parasitic array	End-node	3–5	Low	Medium (low-cost directional control)	Smart buoys, compact sensor platforms
[79,94]	Mechanically steered	Gateway/Ship node	4–6	Moderate	High (robust under motion, coarse alignment)	Vessel-mounted tracking systems
[94,103]	Hybrid Electro-Mechanical	Gateway/Ship node	5–8	Moderate	High (adaptive across wide marine dynamics)	Mobile relays with dynamic coverage need

Units: dB = decibel.

**Table 7 sensors-25-06110-t007:** Research gaps and future directions in maritime adaptive antennas.

Article	Research Gap	Details	Future Direction
[22,31]	Standardization gaps	LoRaWAN lacks protocol support for adaptive features like beamforming; it risks non-compliance with strict antenna limits.	Update LoRaWAN standards to allow adaptive antenna integration.
[36,49,103]	Hardware/Software co-design	Real-time synchronization of hardware and software is complex and power-intensive.	Develop efficient low-power co-design frameworks.
[35,61]	Environmental variability	Vessel movement and sea-surface reflections cause fading; most current studies remain limited to simulations.	Carry out sea trials, refine channel models, and develop adaptive algorithms suited to maritime conditions.
[104,105]	AI and data availability	Few open maritime datasets exist, and most AI models are too computationally heavy for real-time use.	Establish open datasets and create lightweight AI methods to enable efficient real-time beamforming.
[46,71,109]	Cross-medium transmission	Air–sea and air–underwater impairments remain underexplored.	Investigate cross-media models for reliable links.
[26,184,185]	RIS/Energy efficiency	Few studies have examined how RIS can be applied in maritime LoRaWAN, despite showing strong potential for improving coverage, energy use, and spectrum efficiency.	Investigate RIS deployment in sea-based LoRaWAN, for example, by integrating RIS on floating buoys to extend coverage, improve link reliability, and enhance energy efficiency.
[87,186]	Interdisciplinary collaboration	Limited cooperation between engineers and maritime sciences slows progress.	Encourage cross-disciplinary collaboration to enable scalable IoT solutions.

## Data Availability

All data supporting this review are contained within the article.

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
