# Peer review of "Adaptive Antenna for Maritime LoRaWAN: A Systematic Review on Performance, Energy Efficiency, and Environmental Resilience"

_sensors, 2025, doi:10.3390/s25196110_

Round 1
Reviewer 1 Report
Comments and Suggestions for Authors
The paper deals with extensive research on deploying adaptive antenna technology in LoRaWAN communication networks in the maritime environment. The article is very extensive, and several chapters mention some information more than once. The article is unnecessarily long and could be significantly shortened.
My main complaint is related to the direction of the article. The writers are trying to describe the deployment of adaptive antenna technology in LoRa, not LoRaWAN communications. All the above information is related to the principle of Point-to-point communication. The writers do not focus on the communication aspects of LoRaWAN technology. In the described methodology, they do not state what configuration of the LoRaWAN network is used for the analysis and research.
The described antenna systems and their deployment will also vary depending on how many users will be in the given network. This fact is not reflected in the article at all.
In the introduction, they mention modes A, B, and C, which differ fundamentally in the amount of data and the method of communication between the end device and the gateway.
Including a typical environment configuration in the article that corresponds to the information provided would be helpful. The typical LoRaWAN infrastructure shown in Figure 1 is inadequate and does not correspond to real-world deployment in a maritime environment.
One of the key features of LoRaWAN technology is the ability to receive simultaneously on multiple gates, which can fundamentally correct the negative aspects reported during transmission above the water surface. These properties are not included in the methodology and research and should be explicitly presented concerning their importance. It is appropriate to provide references to research that covers this area.
Another important transmission aspect is using the 433 MHz band, which the LoRaWAN standard allows and should be available in most countries worldwide. Such a low frequency can significantly improve communication reliability for specific applications.
The introductory part describes the parameters of transmission technologies, but does not state what applications and requirements are placed on the LoRaWAN network in the maritime environment. It is appropriate to indicate the need for continuous transmission or transmission according to messages (acknowledged, unacknowledged) or the required response for the application. It should be emphasized here that individual SFs in different modes (A, C) specify maximum message sizes from 50 B upwards.
I did not notice a reference to the LoRaWAN standard in the article.
In Table 1, Sigfox technology is now dead; it would be appropriate to mention another alternative. In addition, with Wi-Fi Halow technology, the range is greater than 1 km and can reach up to units of km. WiMAX technology is also dead worldwide today and has been replaced by private networks of LTE and now 5G SA technologies.
All images and formulas throughout the article are small and difficult to read.
I didn't notice the description of the ToA (Time on Air) abbreviation in the article.
Author Response
Comment 1:
“The article is very extensive, and several chapters mention some information more than once. The article is unnecessarily long and could be significantly shortened.”
Response 1:
Thank you for pointing this out. We agree with this comment. Therefore, we have shortened the manuscript by removing redundant explanations and consolidating overlapping content. Specifically, the repeated description of LoRa modulation parameters in the Introduction and Section 2 has been merged into one concise passage, and overlapping discussion of maritime propagation models has been streamlined. In addition, we revised by condensing background details and presenting only the essential context, Section 4 in page 16-19 by summarizing repetitive technical explanations, and Section 5 in page 19-23 by reducing overlapping interpretations of results. These changes have improved readability and ensured a more concise presentation throughout the manuscript.
Comment 2:
“The writers are trying to describe the deployment of adaptive antenna technology in LoRa, not LoRaWAN communications. All the above information is related to the principle of Point-to-point communication. The writers do not focus on the communication aspects of LoRaWAN technology.”
Response 2:
We appreciate this comment. We revised the paper to focus on LoRaWAN system-level aspects rather than point-to-point LoRa. The background now emphasizes LoRaWAN’s star-of-stars architecture, Classes A/B/C, and multi-gateway diversity.
- Change location: Page 3, LoRaWAN Architecture Overview 2.1, Paragraph 1, Lines 103-109
- Updated text:
“The protocol defines three device classes to meet different application needs. Class A is the most energy-conscious: devices only open a receive window after sending, which works well for event-driven uses such as water-quality buoys or emergency alerts. Class B introduces scheduled listening periods, reducing latency and making it more useful for coordination tasks like fleet management. Class C keeps the receiver permanently open, consuming more power but enabling real-time services such as ship tracking, collision avoidance, or remote control [25], [26]. LoRaWAN networks follow a star-of-stars architecture. End devices placed on ships, buoys, or coastal stations collect data such as tide levels, vessel positions, or fishing activity and send it with very low energy use [27].”
Comment 3:
“In the described methodology, they do not state what configuration of the LoRaWAN network is used for the analysis and research.”
Response 3:
We agree. A new subsection has been added to describe the LoRaWAN network configuration assumed in our methodology, including gateway placement, device classes, and frequency bands.
- Change location: Page 5, Section 2,
- Updated text for subsection:
“2.5. LoRaWAN Network Configuration.”
Comment 4:
“The described antenna systems and their deployment will also vary depending on how many users will be in the given network. This fact is not reflected in the article at all.”
Response 4:
We agree. The background section has been expanded to analyze how user density impacts antenna deployment and LoRaWAN scalability in maritime environments.
- Change location: Page 5, Section 2.5, Paragraph 3, Lines 195–202.
- Updated text:
“Deployment scale is another factor that shapes antenna design. For a single buoy or vessel operating far offshore, the priority is to reach the gateway, so high-gain directional antennas are often the best fit. But in crowded zones such as fishing areas, ports, or along coastal infrastructure, the problem shifts. Here, interference between many nodes can reduce reliability, and adaptive or beam-steering antennas become more effective in keeping the network stable. Past studies on LoRaWAN scalability show that when the number of devices rises, packet delivery ratios drop unless new gateways are carefully placed to share the load [18],[50]. This makes it clear that antenna strategies must account not only for range but also for user density”
Comment 5:
“In the introduction, they mention modes A, B, and C… It should be emphasized here that individual SFs in different modes specify maximum message sizes from 50 B upwards.”
Response 5:
We agree. The Introduction has been updated to link Classes A, B, and C to maritime applications and to mention the effect of spreading factors on message size.
- Change location: Page 2, Introduction, Paragraph 4, Lines 70–80.
- Updated text:
“Among all traffic types, the most demanding are alerts, as they rely on highly reliable delivery supported by acknowledgments to ensure the message is received. These needs determine how frequently devices transmit, how much delay is acceptable, and the payload size, which in LoRaWAN ranges from 51 to 243 bytes depending on the spreading factor (SF) [20],[21],[22].”
Comment 6:
“Including a typical environment configuration in the article that corresponds to the information provided would be helpful. The typical LoRaWAN infrastructure shown in Figure 1 is inadequate and does not correspond to real-world deployment in a maritime environment.”
Response 6:
We agree. Figure 1 has been redrawn to reflect a realistic maritime LoRaWAN infrastructure, including ships, buoys, shore gateways, port authority gateways.
- Change location: Page 3, Figure 1.
- Updated figure caption:
“Figure 1. Maritime LoRaWAN architecture.”
Comment 7:
“One of the key features of LoRaWAN technology is the ability to receive simultaneously on multiple gates… These properties are not included in the methodology and research.”
Response 7:
We agree. Multi-gateway reception section has been added, with supporting references.
- Change location: Page 4, Section 2.3, Lines 146.
- Updated text:
“2.3. Multi-Gateway Reception in LoRaWAN”
Comment 8:
“Another important transmission aspect is using the 433 MHz band… Such a low frequency can significantly improve communication reliability.”
Response 8:
We thank the reviewer. A discussion of the 433 MHz band has been added.
- Change location: Page 5, Section 2.5, Line 183-190.
- Updated text:
“This study explicitly considers a LoRaWAN deployment rather than isolated point-to-point LoRa links. The network is assumed to operate primarily in the EU868 MHz band, while also discussing the 433 MHz band as an alternative for extended maritime coverage in line with the LoRaWAN specification” - Change location: Page 21, Section 5.3.3, Line 724-725.
- Updated text:
“LoRaWAN operates in unlicensed bands (433, 868, 915 MHz), with default BW at 125 kHz but extendable up to 500 kHz [35]”.
Comment 9:
“The introductory part does not state what applications and requirements are placed on the LoRaWAN network in the maritime environment.”
Response 9:
We agree. Maritime applications and requirements were added to the Introduction.
- Change location: Page 2, Introduction, Paragraph 4, Lines 70–75.
- Updated text:
“The requirements of maritime IoT are quite different from those of land-based networks. LoRaWAN applications in maritime settings such as vessel tracking, environmental monitoring, and emergency alerts must balance continuous and event-based transmissions while accounting for acknowledgements, latency, and payload limits.”
Comment 10:
“I did not notice a reference to the LoRaWAN standard in the article.”
Response 10:
We appreciate the reviewer’s observation. We have now explicitly cited the official LoRaWAN specifications to strengthen the manuscript. Specifically, we added references to the LoRa Alliance. LoRaWAN® Specification v1.1; LoRa Alliance [22] and the LoRaWAN End Device & Network Server Interoperability Test Plan v1.0.0 [185], which represent the current standard documents issued by the LoRa Alliance. These references ensure that our discussion is aligned with the official LoRaWAN standardization framework.
Comment 11:
“In Table 1, Sigfox technology is now dead… WiMAX technology is also dead worldwide today and has been replaced by LTE and 5G SA.”
Response 11:
We agree. Table 1 was updated: Sigfox and WiMAX were removed, LTE-M and 5G SA private networks were added, and Wi-Fi HaLow range corrected.
- Change location: Page 5, Table 1.
Comment 12:
“All images and formulas throughout the article are small and difficult to read.”
Response 12:
We thank the reviewer. All figures and formulas were reformatted in higher resolution with larger fonts.
- Change location: All figures have been updated eg. Figure 1 in Page 3 redrawn
Comment 13:
“I didn’t notice the description of ToA (Time on Air) abbreviation in the article.”
Response 13:
We agree. ToA is now defined at its first mention.
- Change location: Page 21, Section 5.3.1, Line 711-712.
- Updated text:
“SF affects both the Time-on-Air (ToA) and the communication range [175],[176]. ToA defines the transmission duration of a LoRaWAN packet and depends on the chosen SF, BW, and CR [14].
English Language Improvement
We also carefully revised the manuscript for English usage, grammar, and readability. The text was edited for clarity, sentence flow, and conciseness. These revisions ensure that the research is expressed more clearly and is accessible to the journal’s readership.

Reviewer 2 Report
Comments and Suggestions for Authors
The paper reviews challenges in maritime LoRaWAN communication (multipath fading, interference, energy limits) and analyzes adaptive antennas to improve signal quality. It systematically compares four antenna types (switched beam, phased array, reconfigurable, AI/ML-based) and their impact on SNR, coverage, and packet loss.
Use the numeric number to present the sections in the following paragraph “The organization of this review is as follows: Section two provides a background and an overview of the LoRaWAN technology. Section three presents the methodological framework used for the selection and synthesis of the literature and formulation of the main research questions. Section four focuses on the basic features of adaptive antenna systems and their propagation challenges faced in marine LoRaWAN implementations. Current adaptive antenna structures and integration with LoRaWAN to enhance performance in a dynamic maritime environment are introduced in section five. Section six provides a detailed discussion and thematic synthesis, framed according to research questions, of the selected studies. Lastly, section seven summarises the survey and discusses the conclusions as well as the important open issues and the potential direction for future work in this field.” instead of two, three, four…..
what is the difference between the other surveys like “Banti K, Karampelia I, Dimakis T, Boulogeorgos AA, Kyriakidis T, Louta M. LoRaWAN communication protocols: A comprehensive survey under an energy efficiency perspective. InTelecom 2022 May 25 (Vol. 3, No. 2, pp. 322-357). MDPI.” “Alkhayyal M, Mostafa A. Recent developments in AI and ML for IoT: A systematic literature review on LoRaWAN energy efficiency and performance optimization. Sensors. 2024 Jul 11;24(14):4482.” and the present survey?
Add a section as related studies that will differentiate the present study from the old.
Figure 4 and 6 are not clear; the caption of Figure 7 is not mentioned, Figure 10 is blurry.
Author Response
Comment 1:
“Use the numeric number to present the sections in the following paragraph … instead of two, three, four…”
Response 1:
Thank you for pointing this out. We agree with this comment. Therefore, we revised the manuscript to use numeric references (e.g., “Section 2” instead of “Section two”) consistently throughout the paper.
- Change location: Page 2, Introduction, Paragraph 6 line 89-94
- Updated text:
“The paper is structured in seven sections. Section 2 introduces LoRaWAN and explains why its features matter in maritime applications. Section 3 describes the review method. Section 4 looks at the main propagation challenges together with antenna fundamentals, while Section 5 discusses design options and their integration into LoRaWAN systems. Section 6 brings the results together in relation to the research questions, and Section 7 closes with open problems and directions for future work.”
Comment 2:
“What is the difference between the other surveys like Banti et al. (2022) and Alkhayyal & Mostafa (2024) and the present survey?”
Response 2:
We agree with this important point. To clarify the novelty of our contribution, we have added a new subsection titled “Related Studies” in the Background section. This subsection contrasts previous surveys with our work. Prior studies mainly focused on LoRaWAN under energy efficiency perspectives or on AI/ML-driven optimizations, while our study uniquely investigates adaptive antenna technologies for maritime LoRaWAN deployments. The discussion highlights how this review systematically synthesizes antenna approaches (switched beam, phased array, reconfigurable, AI/ML-based) to address multipath fading, mobility, and interference in marine environments.
- Change location: Page 3, Section 2 (Background)
- Manuscript Changes:
Added Section “2.2. Related Studies”. Also cited and discussed works by Banti et al. (2022), Alkhayyal & Mostafa (2024), and other key reviews.
Comment 3:
“Add a section as related studies that will differentiate the present study from the old.”
Response 3:
We agree. Accordingly, we added a new subsection titled “Related Studies” in Section 2 to differentiate this work from previous surveys and emphasize our unique contribution in maritime LoRaWAN adaptive antennas.
- Change location: Page 3, Section 2 (Background)
Added Section “2.2. Related Studies”.
Comment 4:
“Figure 4 and 6 are not clear; the caption of Figure 7 is not mentioned; Figure 10 is blurry.”
Response 4:
We thank the reviewer for pointing this out. All figures have been redrawn and reformatted for clarity and journal consistency: Figure 4 and Figure 6 were redrawn with higher resolution and larger fonts. Figure 7 caption was added and aligned with journal requirements. Figure 10 was redrawn in high resolution, cleaned layout, and improved color contrast for readability.
- Change location: Page 15, Figure 4; Page 17, Figure 6; Page 21, Figure 7; Page 26, Figure 10.
- Manuscript Changes
- Figure 4 and Figure 6 were redrawn with higher resolution and larger fonts.
- Figure 7 caption was added and aligned with journal requirements. “Figure 7. LoRa protocol stack: PHY and MAC layers, adapted from [171].”
- Figure 10 was redrawn in high resolution, cleaned layout, and improved color contrast for readability.
English Language Improvement
We also carefully revised the manuscript for English usage, grammar, and readability. The text was edited for clarity, sentence flow, and conciseness. These revisions ensure that the research is expressed more clearly and is accessible to the journal’s readership.

Reviewer 3 Report
Comments and Suggestions for Authors
The paper presents a survey on adaptive antenna for maritime LoRaWAN. The reviewer has the following comments.
- The introduction should describe your contributions in presenting this survey better or in a more specific way. Also, the section could give a clear structure of the paper to inform the key sections of the paper and how they are linked together. The paper should also focus on presenting papers related to maritime LoRaWAN. There are many survey papers about LoRaWan. Your focus may bring some interesting materials to the readers.
- Section 2 should have more specific title than Background. Also, as the paper is about maritime LoRaWAN, the section should focus on the background of maritime LoRaWAN instead of the general LoRaWAN. Figure 1 should also focus on a maritime LoRaWAN. The requirement subsection (2.3) should consider the requirements for cross-medium transmissions. I suggest that you can refer to the paper: dual RIS-aided parallel intelligence surface for IoAMVSs: a co-design approach for 3C problems. RISs form a key technology for the next generation wireless communications and IoT networks. The paper should discuss how RISs may support maritime IoT/LoRaWAN. Particularly, there are quite a few studies on RISs’ energy harvesting and data transmissions that are designed for IoT devices/applications. For example, minimizing BER without CSI for dynamic RIS-assisted wireless broadcast communication systems.
- The quality of figures in the paper should be improved. Fonts and curves in these figures are illegible. Please make sure that Figure 5 presents correct information.
- More critical analysis should be presented. Also, the future research directions are what readers want to read.
Author Response
Comment 1:
“The introduction should describe your contributions in presenting this survey better or in a more specific way. Also, the section could give a clear structure of the paper to inform the key sections of the paper and how they are linked together. The paper should also focus on presenting papers related to maritime LoRaWAN. There are many survey papers about LoRaWAN. Your focus may bring some interesting materials to the readers.”
Response 1:
Thank you for this valuable comment. We revised the Introduction to: (i) clearly state the specific contributions of this survey, (ii) provide a clear roadmap of the paper, and (iii) emphasize that our review uniquely focuses on maritime LoRaWAN, differentiating it from previous general LoRaWAN surveys.
- Change location: Page 2–3, Introduction, Paragraphs 5–6, Lines 81–94.
- Updated text:
“ This review looks at how adaptive antenna designs could be uniquely applied to maritime LoRaWAN. We grouped existing antennas into switched-beam, phased-array, reconfigurable, and AI/ML-enabled approaches, then examine their reported benefits for coverage, reliability, and energy efficiency in open-water conditions. The review also has the following contributions: (i) it examines propagation challenges and adaptive antenna solutions in maritime LoRaWAN environments; (ii) it classifies antenna types and evaluates their strengths and limitations; (iii) it synthesizes maritime application requirements; and (iv) it highlights open challenges and future research directions.”
“The paper is structured in seven sections. Section 2 introduces LoRaWAN and ex-plains why its features matter in maritime applications. Section 3 describes the review method. Section 4 looks at the main propagation challenges together with antenna fundamentals, while Section 5 discusses design options and their integration into LoRaWAN systems. Section 6 brings the results together in relation to the research questions, and Section 7 closes with open problems and directions for future work.”
Comment 2:
“Section 2 should have more specific title than Background. Also, as the paper is about maritime LoRaWAN, the section should focus on the background of maritime LoRaWAN instead of the general LoRaWAN. Figure 1 should also focus on a maritime LoRaWAN. The requirement subsection (2.3) should consider the requirements for cross-medium transmissions. I suggest that you can refer to the paper: dual RIS-aided parallel intelligence surface for IoAMVSs: a co-design approach for 3C problems. RISs form a key technology for the next generation wireless communications and IoT networks. The paper should discuss how RISs may support maritime IoT/LoRaWAN. Particularly, there are quite a few studies on RISs’ energy harvesting and data transmissions that are designed for IoT devices/applications. For example, minimizing BER without CSI for dynamic RIS-assisted wireless broadcast communication systems.”
Response 2:
We appreciate this constructive suggestion. Section 2 has been retitled “Background on Maritime LoRaWAN”. The content now focuses on maritime-specific challenges. Figure 1 was redrawn to show a maritime LoRaWAN deployment (buoys, vessels, shore gateways). Subsection 2.3 has been expanded with a new paragraph on cross-medium transmission requirements (air–sea–land), supported with recent references. We also added discussion on Reconfigurable Intelligent Surfaces (RISs), highlighting their potential for maritime IoT/LoRaWAN, including energy harvesting and BER minimization without full CSI.
- Change location: Page 3-4, Section 2 Maritime “Background on Maritime LoRaWAN”; Page 4, Subsection 2.4, Lines 175–181; Page 10, new RIS discussion.
- Updated text:
Subsection “2.4: Maritime Specific Requirements.”
… “A further complication is that some applications must link devices across different media. LoRaWAN works well in the air and over water, but signals fade quickly below the surface, which prevents direct underwater communication [46],[47]. Hybrid solutions through mixing RF with acoustic or optical links, or even using UAVs as relays, are now being tested to bridge the gap between surface nodes and underwater assets [48]. Studies agree that handling these cross-media transitions will be key to building robust and scalable maritime IoT systems [49].”
New subsection: “2.8. Reconfigurable Intelligent Surfaces (RIS) for Maritime LoRaWAN”
Comment 3:
“The quality of figures in the paper should be improved. Fonts and curves in these figures are illegible. Please make sure that Figure 5 presents correct information.”
Response 3:
We agree with this observation. All figures have been redrawn with improved resolution, larger fonts, and clearer curves. Figure 5 was carefully verified and corrected to ensure accurate information.
- Change location: Page 3, Figures 1; Page 7-8, Figures 2-3; Page 16, Figure 4. Page 17, Figures 5; Page 18, Figures 6; Page 21, Figures 7; Page 24, Figures 8; Page 25, Figures 9; Page 26, Figures 10;
Comment 4:
“More critical analysis should be presented. Also, the future research directions are what readers want to read.”
Response 4:
Thank you for this suggestion. We strengthened the Results and Discussion section with more critical analysis of adaptive antenna types (switched-beam, phased arrays, reconfigurable, AI/ML-based) for maritime LoRaWAN, comparing their advantages and limitations. The Conclusion has been expanded to include detailed future research directions, including RIS integration, AI/ML for adaptive beamforming, and low-frequency operation (433 MHz) for improved maritime coverage.
- Change location: Page 29, Section 6 (Results and Discussion); Subsection “6.7. Synthesis of the Findings” Page 29, Section 7 (Conclusion). Lines 900-950
English Language Improvement
We also carefully revised the manuscript for English usage, grammar, and readability. The text was edited for clarity, sentence flow, and conciseness. These revisions ensure that the research is expressed more clearly and is accessible to the journal’s readership.

Round 2
Reviewer 1 Report
Comments and Suggestions for Authors
The article has been expanded and supplemented to include most of the above comments. About the information given on smart antennas, the article lacks an explicit description and diagram of how this kind of antenna could be connected to a LoRaWAN device and who will be in charge of its control. Not all antenna systems mentioned operate independently, similar to the PAS described in Section 4.3.1.
On lines 189-190, the statement about the duty cycle in the EU needs to be corrected, which is not just 1% across the board, but depending on the channel used, it varies from 0.1 to 10%. Similar information is given on line 719.
There are still many typographical problems in the article. Most of the figures are illegible and unnecessarily large, see 4, 7, and others. Some equations are also poorly planted; see the equation in line 229. Given the size of the document, it is necessary to carry out a complete typographical revision before publication.
Author Response
We thank Reviewer 1 for their additional feedback, which has further improved the manuscript. Our detailed responses are provided below.
Comment 1: “The article lacks an explicit description and diagram of how smart antennas could be connected to a LoRaWAN device and who will be in charge of its control. Not all antenna systems mentioned operate independently, similar to the PAS described in Section 4.3.1.”
Response: We agree and have added a new subsection, “Adaptive Antenna Integration in LoRaWAN”, which explicitly explains how adaptive antennas are incorporated at the device, gateway, and server levels. To support this, a new schematic has been included as Figure 4. This figure shows the hierarchical control structure, with switched-beam antennas managed by the device MCU and more complex phased-array/AI-based antennas managed at the gateway with optional server coordination.
Manuscript Changes: New subsection “2.7.5. Adaptive Antenna Integration in LoRaWAN” added; Figure 4 added in Page 10
Comment 2: “On lines 189–190, the statement about the duty cycle in the EU needs to be corrected. It is not always 1%, but varies between 0.1% and 10% depending on the channel. Similar issue on line 719.”
Response: This has been corrected. The manuscript now specifies that the EU duty cycle varies between 0.1% and 10%, depending on the frequency sub-band, consistent with LoRaWAN Regional Parameters.
Manuscript Changes: Corrections made in Section 2.5 LoRaWAN Network Configuration Page 5 Lines 190-192;
“In the EU868 MHz band, duty-cycle limits range between 0.1% and 10% depending on the sub-band [22],[43], and must be taken into account when designing maritime LoRaWAN deployments to balance compliance and throughput.”
Corrections made in Section 5.3.2. Transmission Power, Page 22, Lines 739-741
“LoRa devices typically operate at 13–20 dBm depending on the frequency band [178]. Tx settings are constrained by duty-cycle regulations (ranging from 0.1% to 10% in the EU868 band), as well as regional rules on ToA, channel hopping, and packet duration [179].”
Comment 3: “There are still many typographical problems. Most figures are illegible and unnecessarily large (see Figures 4, 7, and others). Some equations are poorly formatted (e.g., line 229). A complete typographical revision is necessary.”
Response: A thorough typographical revision has been performed. Figures were redrawn in high resolution font and appropriately resized for readability. Equations were reformatted according to journal guidelines. Specifically, the PRISMA flow diagram was resized (Figure 4 which is now Figure 5), adapted from the PRISMA template), and Figure 7 which is now Figure 8 has been updated with a high-resolution redraw to remove blurriness.
Manuscript Changes: Figures 5, 8, in pages 16 and 22 respectively and others revised and resized; equations reformatted; full typographical check completed.

Reviewer 2 Report
Comments and Suggestions for Authors
Figure 7 still blurry. It needs to update.
Author Response
We sincerely thank Reviewer 2 for their constructive follow-up feedback.
Comment: “Figure 7 is still blurry. It needs to update.”
Response: We fully acknowledge this concern and have replaced the figure (now Figure 8) with a newly updated, high-resolution version, eliminating the blurriness of the earlier version.
Manuscript Changes:
Figure 8.LoRa protocol stack: PHY and MAC layers, adapted from [171] Page 22
Reviewer 3 Report
Comments and Suggestions for Authors
I have no further comments.
Author Response
We thank Reviewer 3 for their time and constructive feedback during the review process. We are pleased that the revised manuscript has addressed their concerns, and we appreciate their confirmation that no further comments remain.